# Combinatorial Sparse PCA Beyond the Spiked Identity Model

**Syamantak Kumar** [1]  **Purnamrita Sarkar** [1]  **Kevin Tian** [1]  **Peiyuan Zhang** [2]

## Abstract

Sparse PCA is one of the most well-studied problems in high-dimensional statistics. In this problem, we are given samples from a distribution with covariance $\boldsymbol{\Sigma}$, whose top eigenvector $\mathbf{v} \in \mathbb{R}^d$ is $s$-sparse. Existing sparse PCA algorithms can be broadly categorized into (1) combinatorial algorithms (e.g., diagonal or elementwise covariance thresholding) and (2) SDP-based algorithms. While combinatorial algorithms are much simpler and more efficient, they are typically analyzed only under the spiked identity model (where $\boldsymbol{\Sigma} \propto \mathbf{I}_d + \gamma \mathbf{v}\mathbf{v}^\top$ for some $\gamma > 0$), whereas SDP-based algorithms require no additional assumptions on $\boldsymbol{\Sigma}$.

We demonstrate explicit counterexample covariances $\boldsymbol{\Sigma}$ against the success of standard combinatorial algorithms for sparse PCA, when moving beyond the spiked identity model. In light of this discrepancy, we give the first combinatorial method for sparse PCA that provably succeeds for general $\boldsymbol{\Sigma}$ using $\mathrm{poly}(s, \log(d))$ samples and $d^2 \cdot \mathrm{poly}(s, \log(d))$ time, by providing a global convergence guarantee on the truncated power method of (Yuan and Zhang, 2013). We provide a natural generalization of our method to recover a vector in a planted sparse subspace. Finally, we evaluate our method on synthetic and real-world sparse PCA datasets.

## 1. Introduction

Principal component analysis (PCA) is a classical and widely-used tool for dimensionality reduction that identifies directions in which the data exhibit the most variance. Interestingly, the classical intuition that the leading eigenvectors of the sample covariance reliably estimate the population-level principal components can fail dramatically in high

[1]UT Austin [2]UW Madison. Correspondence to: Kevin Tian <kjtian@cs.utexas.edu>.

*Proceedings of the $43^{rd}$ International Conference on Machine Learning*, Seoul, South Korea. PMLR 306, 2026. Copyright 2026 by the author(s).

dimensions (Paul, 2007; Jung and Marron, 2009; Johnstone and Lu, 2009a). Even when $\frac{d}{n} \to c > 0$ with (the dimension and sample size) $d, n \to \infty$, and the population covariance has a single spiked diagonal entry, the BBP phase transition (Baik et al., 2005) identifies a sharp *eigenvalue* threshold at which the top empirical eigenvalue separates from the Marchenko–Pastur bulk; below this threshold no outlier emerges, and the leading empirical eigenvector is asymptotically uninformative, exhibiting vanishing correlation with the population eigenvector (Paul, 2007).

Sparse PCA addresses this inconsistency by positing additional structure: that the leading eigenvector $\mathbf{v}_1$ of the population covariance is $s$-sparse, reflecting that only a small subset of variables drives the dominant variation. The canonical finite-sample optimization problem is

$$\max_{\substack{\|\mathbf{v}\|_2=1 \\ |\mathrm{supp}(\mathbf{v})| \leq s}} \mathbf{v}^\top \widehat{\boldsymbol{\Sigma}} \mathbf{v}, \tag{1}$$

where $\widehat{\boldsymbol{\Sigma}}$ is the sample covariance. While sparsity can restore statistical consistency and yield sample complexities $\approx s \log d$ rather than $\Omega(d)$, (1) is nonconvex and NP-hard in general (Magdon-Ismail, 2015), motivating the study of polynomial-time approximation algorithms (d'Aspremont et al., 2008; Amini and Wainwright, 2008; Johnstone and Lu, 2009b; Vu and Lei, 2012).

**Models of sparse PCA.** Existing sparse PCA algorithms can be grouped by the type of covariance model assumed. The first is the well-known spiked identity covariance model.[1]

**Model 1** (Spiked identity model). *Fix* $(s, d, n) \in \mathbb{N}^3$ *and* $\gamma \in (0, 1)$. *In the* spiked identity model, *there is an unknown unit* $\mathbf{v} \in \mathbb{R}^d$ *with* $\mathrm{nnz}(\mathbf{v}) \leq s$. *We obtain* $\{\mathbf{x}_i\}_{i \in [n]} \sim_{\text{i.i.d.}} \mathcal{D}$, *a sub-Gaussian distribution with covariance* $\boldsymbol{\Sigma} = 0.9(\mathbf{I}_d - \mathbf{v}\mathbf{v}^\top) + \mathbf{v}\mathbf{v}^\top$.

Model 1 has inspired elegant yet simple combinatorial principles for solving sparse PCA. Notable examples are the diagonal thresholding method (Johnstone and Lu, 2009b; Amini and Wainwright, 2008), which simply keeps the top $s$ elements of the marginal variances (the diagonal of the empirical covariance). Another line of work thresholds the sample covariance matrix itself to establish consistency (Bickel

---

[1]For definitions and notation used throughout, see Section 2.

and Levina, 2009; Deshpande and Montanari, 2016; Wainwright, 2019; Novikov, 2023). While these combinatorial methods are computationally light and interpretable, they typically require structural assumptions, e.g., working under the spiked identity model (Model 1).

Our focus is combinatorial algorithms for the sparse PCA problem in full generality (Model 2), i.e., assuming only that the top eigenvector of $\mathbf{\Sigma}$ is sparse.

**Model 2** (General model). *Fix* $(s, d, n) \in \mathbb{N}^3$, *and* $\gamma \in (0, 1)$. *In the* general model, *there is an unknown unit* $\mathbf{v} \in \mathbb{R}^d$ *with* $\mathrm{nnz}(\mathbf{v}) \leq s$, *the top eigenvector of an unknown* $\mathbf{\Sigma} \in \mathbb{S}_{\succeq \mathbf{0}}^{d \times d}$ *with* $\lambda_2(\mathbf{\Sigma}) \leq 0.9\lambda_1(\mathbf{\Sigma})$. *We obtain* $\{\mathbf{x}_i\}_{i \in [n]} \sim_{\mathrm{i.i.d.}} \mathcal{D}$, *a sub-Gaussian distribution with covariance* $\mathbf{\Sigma}$.

In both Models 1 and 2, we set the "gap" parameter separating the quality of the ($s$-sparse) top eigenvector $\mathbf{v}$ from any alternative, to be $0.1$ for simplicity; our main results are derived under a generalization to an arbitrary gap $\gamma > 0$ (cf. Model 3). Model 1 is a special case of Model 2 where the covariance outside the $\mathbf{v}$ subspace is a (perfectly-spherical) multiple of $\mathbf{I}_d - \mathbf{v}\mathbf{v}^\top$ exactly. Thus, Model 2 generalizes Model 1 by making only the minimal well-posedness assumptions for sparse PCA: that $\lambda_2(\mathbf{\Sigma}) \leq 0.9\lambda_1(\mathbf{\Sigma})$, and that $\mathbf{\Sigma}$ has a sparse top eigenvector.

Our starting point is the surprising observation that under Model 2, standard combinatorial algorithms (many of which have provable recovery guarantees under Model 1) fail with constant probability under the conjectured minimum $n \gtrsim s^2 \log d$ for polynomial-time algorithms to exist (Berthet and Rigollet, 2013). Indeed, in Appendix B, we give counterexamples against diagonal thresholding, covariance thresholding, and a planted clique-style greedy correlation ranking suggested by (Błasiok et al., 2024). These counterexamples rule out robustness of typical $\approx d^2 \cdot \mathrm{poly}(s \log d)$-time combinatorial methods beyond Model 1.

Under Model 2, most provable methods use heavy hammers from convex programming, most prominently semidefinite programming (SDP)-based formulations (d'Aspremont et al., 2008; Amini and Wainwright, 2008; Vu and Lei, 2012; Berthet and Rigollet, 2013; Wang and Lu, 2016; d'Aspremont et al., 2008; Vu et al., 2013; Sriperumbudur et al., 2007; Zou and Xue, 2018; Dey et al., 2017; Amini and Wainwright, 2008). These algorithms are computationally burdensome and memory-intensive in high dimensions: for the standard $\ell_1$-constrained sparse PCA SDP with $\approx d^2$ constraints and a $d \times d$ decision variable, state-of-the-art SDP solvers run in time $\Omega(d^{2\omega}) \subseteq \Omega(d^{4.5})$ in theory (where $\omega > 2.371$ denotes the current matrix multiplication constant) (Huang et al., 2022; Alman et al., 2025), and may incur even larger overheads in practice.

Most SDP-free algorithms for the general covariance model are spectral in nature. The truncated power method as proposed and analyzed in (Yuan and Zhang, 2013) was previously only known to succeed under a suitable local initialization. Recently, (Qiu et al., 2023) proposed a single-pass streaming algorithm and claim it is the first to provably reach the global optimum without local initialization under a general $\mathbf{\Sigma}$. The method requires $O(d^2)$ memory, $O(nd^2)$ time, but achieves error $O(\frac{d^2}{\sqrt{n}})$, so it requires e.g., $\mathrm{poly}(d)$ samples for constant error. A follow-up work by (Kumar and Sarkar, 2024) introduced a computationally lightweight, linear-time streaming algorithm, though its statistical guarantees again require a larger sample size than is information-theoretically optimal.

This state of affairs begs our central question.

*Is there a lightweight combinatorial method that*
*solves sparse PCA under Model 2?*

Concretely, our goal is to design an algorithm succeeding under Model 2 with $\mathrm{poly}(s \log(d))$ samples and $d^2 \cdot \mathrm{poly}(s \log(d))$ time, competitive with existing combinatorial methods for Model 1. Despite the rich body of work on sparse PCA and many attempts to answer the above question, this fundamental problem has remained open.

**The semi-random lens.** Model 2 has connections to a recent line of work on *semi-random models* from theoretical computer science, whose goal is to investigate the robustness of statistical algorithms to their modeling assumptions. Briefly, a (monotone) semi-random model changes a standard statistical model by strengthening the desired signal, in a way that breaks an original modeling assumption. The underlying philosophy is that if a statistical algorithm fails under such a "helpful" modification, then the design of the algorithm was overfit to the model.

Our setting is reflective of this philosophy, as compared to Model 1, Model 2 can only amplify the signal in the $\mathbf{v}$ direction: it posits that all other eigenvalues are *at most* $0.9\lambda_1(\mathbf{\Sigma})$, whereas Model 1 posits that they are all *exactly* $0.9\lambda_1(\mathbf{\Sigma})$. It is thus potentially troublesome from a robustness standpoint that existing combinatorial heuristics fail under a more general model, that can only amplify the signal towards $\mathbf{v}$ from an information-theoretic point of view.

The semi-random philosophy has been applied to a wide range of statistical problems, starting from seminal work by (Blum and Spencer, 1995; Feige and Kilian, 2001) on graph coloring. We are particularly inspired by a line of recent work investigating the brittleness of *lightweight, fast algorithms* to semi-random models (as opposed to less efficient convex programming counterparts). This investigation has been applied to (sparse) linear regression (Cheng and Ge, 2018; Jambulapati et al., 2023; Kelner et al., 2023; Gao and Cheng, 2023; Kelner et al., 2024), learning (Blum et al., 2024; Chandrasekaran et al., 2024), and clustering

(Bhaskara et al., 2024; Błasiok et al., 2024). Notably, the work of (Błasiok et al., 2024) designed fast combinatorial algorithms for a semi-random variant of the planted clique problem (building upon (Buhai et al., 2023), which used SDP machinery). Due to the intimate connection between planted clique and sparse PCA (Berthet and Rigollet, 2013), we were directly motivated by the development in (Błasiok et al., 2024), and we analyze the combinatorial heuristic in that paper as it applies to our problem in Appendix B.

### 1.1. Contributions

We now outline our primary contributions.

**Restarted truncated power method.** We provide a global convergence analysis on a modification of the truncated power method, proposed by (Yuan and Zhang, 2013) as a combinatorial heuristic for sparse PCA, but previously only analyzed under a suitable local initialization. The algorithm applies truncation to the iterates of the power method, keeping the largest $r$ coordinates by magnitude.

---

**Algorithm 1:** $\mathsf{RTPM}(\{\mathbf{x}_i\}_{i \in [n]}, r, T)$

1 **Input:** Dataset $\{\mathbf{x}_i \in \mathbb{R}^d\}_{i \in [n]}$, sparsity hyperparameter $r \in [d]$, iteration count $T \in \mathbb{N}$
2 $\widehat{\boldsymbol{\Sigma}} \leftarrow \frac{1}{n} \sum_{i \in [n]} \mathbf{x}_i \mathbf{x}_i^\top$
3 **for** $i \in [d]$ **do**
4     $\mathbf{u}_0^{(i)} \leftarrow \mathbf{e}_i$
5     **for** $t \in [T]$ **do**
6        $\mathbf{u}_t^{(i)} \leftarrow \mathrm{top}_r(\widehat{\boldsymbol{\Sigma}} \mathbf{u}_{t-1}^{(i)}) \cdot \|\mathrm{top}_r(\widehat{\boldsymbol{\Sigma}} \mathbf{u}_{t-1}^{(i)})\|_2^{-1}$
7     **end**
8 **end**
9 **return** $\mathbf{u}_T^{(i)}$ where $i \in \arg\max_{i \in [d]} \langle \mathbf{u}_T^{(i)}, \widehat{\boldsymbol{\Sigma}} \mathbf{u}_T^{(i)} \rangle$

---

Our modification is described in Algorithm 1, and we provide the following convergence guarantee.

**Theorem 1** (Informal, see Theorem 2). *Let $\delta \in (0, 1)$, and under Model 2, assume that*

$$n = \Omega\left(s^3 \log\left(\frac{d}{\delta}\right)\right), \quad r = \Omega\left(s^2\right), \quad T = \Omega\left(\log s\right)$$

*for appropriate constants. Then, in time $O\left(nd^2 T\right)$, Algorithm 1 returns an $r$-sparse unit vector $\mathbf{u}$ such that with probability at least $1 - \delta$, $\langle \mathbf{v}, \mathbf{u} \rangle^2 \geq \frac{9}{10}$.*

We remark that the output can be made exactly $s$-sparse via a final truncation step (Lemma 8), at constant overhead to the final approximation error. Our formal theorem statement, Theorem 2, is stated for an arbitrary gap $\gamma$ and final correlation $\Delta$ to $\mathbf{v}$ (with dependences explicitly stated); Theorem 1, specializes the statement to $\Delta = \gamma = \frac{1}{10}$. This marks the first algorithm that achieves our desired efficiency targets

of poly$(s \log(d))$ samples and $d^2 \cdot$ poly$(s \log(d))$ time, for sparse PCA under Model 2, improving upon SDP-based counterparts by $\Omega(d^{2.5})$ in the runtime. We do note that our sample complexity scales as $\approx s^3 \log(d)$, and leave it as an exciting open problem to achieve a similar runtime using the conjectured information-theoretically optimal $\approx s^2 \log(d)$ samples, as the slower SDP-based methods require. Furthermore, we note that the dependence on $\Delta, \gamma$, stated in Theorem 2, is suboptimal, and we leave improving this dependence as an interesting future direction. Empirical experiments in Appendix F suggest that this dependence can be significantly milder in practice.

To prove our result, we begin by observing that initializing the truncated power method of (Yuan and Zhang, 2013) with a standard basis vector indicating the largest coordinate of $\mathbf{v}$ achieves poly$(\frac{1}{s})$ correlation, making it a suitable local initialization. Thus, we simply try all possible initializations in Theorem 2, a *restarted* truncated power method. We complement this observation with *oversampling the support*, i.e., choosing a truncation parameter $r \gg s$, to account for the low correlation in early iterations. Our final analysis requires care in converting between notions of PCA, as the (Yuan and Zhang, 2013) argument proceeds by analyzing the correlation with $\mathbf{v}$, whereas the selection step in Line 9 cannot measure this quantity and instead uses empirical quadratic forms. Combining these insights yields our main result, Theorem 2, for sparse PCA under Model 2.

**Towards sparse subspace recovery.** We complement our results by considering the more general problem of recovering a $k$-dimensional leading eigenspace, whose components all share a support of size $s$. Existing SDP-based algorithms for sparse (1)-PCA also solve this more general problem of sparse subspace recovery, as we recall for completeness in Appendix E (following (Vu et al., 2013)).

Several works in the literature, (Mackey, 2008; Gataric et al., 2020) have proposed a deflation-based strategy for reducing the sparse $k$-PCA problem to iteratively solving sparse 1-PCA problem on a deflated matrix (the original target, with all previously-learned approximate principal components projected out). We provide pseudocode for such a reduction in Algorithm 2. Recently, the approximation tolerance of this strategy was rigorously analyzed in the non-sparse PCA setting by (Jambulapati et al., 2024). Unfortunately, despite decades of research, to our knowledge no theoretical analysis of deflation-based $k$-sparse PCA reductions currently exists under Model 3, the natural generalization of Model 2.

We investigate this lack of theoretical guarantees for deflation-based sparse PCA methods in Section 3.2. We provide a formal barrier against such reductions for sparse PCA: namely, that Model 3 does not reduce to an instance of itself after deflation, under mild approximation error. Our counterexample (Lemma 9) constructs $\boldsymbol{\Sigma} \in \mathbb{S}_{\succeq \mathbf{0}}^{d \times d}$ with a

leading top-2 eigenspace supported on just $s = 2$ coordinates, such that after projecting out a 3-sparse vector that is arbitrarily well-correlated with the top eigenvector, the deflated matrix suddenly has a fully-dense top eigenvector. Thus, sparse PCA methods no longer apply to the residual problem. Interestingly, our counterexample leverages the generality of Model 2, again highlighting the potential brittleness of existing sparse PCA frameworks.

Nevertheless, we show in Theorem 2 that under Model 3, RTPM does successfully approximate a single sparse eigenvector in the leading eigenspace. This opens the door to its use in any potential future deflation-based sparse PCA frameworks that can bypass the barrier in Lemma 9.

### 1.2. Related work

In addition to the work cited in the introduction, we overview more related work on sparse PCA in this section, as the literature has studied several variants of the problem.

**Existing algorithms for Model 1.** A large body of work (Ma, 2013; Cai et al., 2013; Yang and Xu, 2015; Wang and Lu, 2016; Bresler et al., 2018; Gataric et al., 2020; Cai et al., 2025) develops algorithms with provable convergence guarantees for the spiked identity model (Model 1) with either 1 or $k$ spikes. The strategies therein range from using marginal variances for providing a warm start, a thresholding-based variant of power method, a reduction to sparse PCA, and using random projections with a deflation-style approach. While these algorithms tend to be lightweight and combinatorial in nature, none provably extend to Model 2 (and indeed, we provide counterexamples to several prominent ones in Appendix B).

**Existing algorithms for Model 2.** As discussed earlier, existing provable algorithms for solving sparse PCA under Model 2 using $\mathrm{poly}(s \log(d))$ samples are based on solving SDPs. Here we outline several computationally-cheaper alternatives in the literature and their guarantees.

Most SDP-free algorithms for the general covariance model are spectral in nature. The truncated power method as proposed and analyzed in (Yuan and Zhang, 2013) was previously only known to succeed under a suitable local initialization. Recently, (Qiu et al., 2023) proposed a single-pass streaming algorithm and claim it is the first to provably reach the global optimum without local initialization under a general $\boldsymbol{\Sigma}$. The method requires $O(d^2)$ memory, $O(nd^2)$ time, but achieves error $O(\frac{d^2}{\sqrt{n}})$, so it requires e.g., $\mathrm{poly}(d)$ samples for constant error. A follow-up work by (Kumar and Sarkar, 2024) introduced a computationally lightweight, linear-time streaming algorithm, though its statistical guarantees again require a larger sample size ($\Omega(\mathrm{poly}(d))$) than is information-theoretically optimal.

**Optimization and approximation viewpoints.** Several works study sparse PCA primarily as a *deterministic* optimization problem by developing exact or approximation algorithms, and geometric characterizations. These do not provide statistical recovery rates as a function of the sample size $n$. Representative examples include (Chowdhury et al., 2021; Bertsimas and Kitane, 2023; Chen and Rohe, 2024; Pia et al., 2025; Li and Xie, 2025).

**Computational-statistical gaps.** Building on the foundational work of (Berthet and Rigollet, 2013; Wang et al., 2016), where a conjectured computational-statistical gap for sparse PCA was shown via a reduction to the planted clique problem, recent work has studied sparse PCA in intermediate computational regimes, including certification hardness for constrained PCA (Bandeira et al., 2020; Potechin and Rajendran, 2022; Ding et al., 2023); closely related computational barriers also appear in sparse CCA (Gao et al., 2017). However, unlike the more fine-grained computational-statistical tradeoffs that our work focuses on, these works primarily examine the gap between the statistical properties of polynomial-time algorithms vs. exponential-time algorithms.

## 2. Preliminaries

**Notation**. For $n \in \mathbb{N}$ we let $[n] := \{i \in \mathbb{N} \mid i \le n\}$. We denote vectors in lowercase boldface letters and matrices in capital boldface letters. We denote the $i^{\text{th}}$ canonical basis vector in $\mathbb{R}^d$ by $\mathbf{e}_i$. For $p \in [1, \infty]$ we let $\|\mathbf{v}\|_p$ to denote the $\ell_p$ norm of $\mathbf{v}$. We use nnz to denote the number of nonzero entries in a vector or matrix, and supp to denote the support (i.e., indices of the nonzero entries). We use $\mathbf{0}_d$ to denote the all-zeroes vector and $\mathbf{1}_d$ to denote the all-ones vector in $\mathbb{R}^d$, and for an event $\mathcal{E}$, we use $\mathbb{1}(\mathcal{E})$ to denote the corresponding 0-1 indicator random variable.

For matrices $\mathbf{A}, \mathbf{B}$ with the same number of rows, we let $\begin{pmatrix} \mathbf{A} & \mathbf{B} \end{pmatrix}$ denote their horizontal concatenation. We let $\mathbf{I}_d$ be the $d \times d$ identity and $\mathbf{0}_{m \times n}$ be the all-zeroes $m \times n$ matrix. We say $\mathbf{V} \in \mathbb{R}^{d \times r}$ is orthonormal if its columns are orthonormal, i.e. $\mathbf{V}^\top \mathbf{V} = \mathbf{I}_r$; note that $d \ge r$ in this case. We call a set of vectors orthonormal if their horizontal concatenation is orthonormal. We let $\mathbb{S}^{d \times d}$ be the set of real symmetric $d \times d$ matrices, which we equip with the Loewner partial ordering $\preceq$ and the Frobenius inner product $\langle \mathbf{M}, \mathbf{N} \rangle = \mathrm{Tr}(\mathbf{MN})$. We let $\mathbb{S}^{d \times d}_{\succeq \mathbf{0}}$ and $\mathbb{S}^{d \times d}_{\succ \mathbf{0}}$ respectively denote the positive semidefinite and positive definite subsets of $\mathbb{S}^{d \times d}$.

We say $\mathbf{A} \in \mathbb{R}^{m \times n}$ has singular value decomposition (SVD) $\mathbf{U \Sigma V}^\top$ if $\mathbf{A} = \mathbf{U \Sigma V}^\top$ has rank-$r$, $\boldsymbol{\Sigma} \in \mathbb{R}^{r \times r}$ is diagonal, and $\mathbf{U} \in \mathbb{R}^{m \times r}$, $\mathbf{V} \in \mathbb{R}^{n \times r}$ are orthonormal. We refer to the $i^{\text{th}}$ largest eigenvalue of $\mathbf{M} \in \mathbb{S}^{d \times d}$ by $\lambda_i(\mathbf{M})$, the corresponding eigenvector by $\mathbf{v}_i(\mathbf{M})$, and the $i^{\text{th}}$ largest

singular value of $\mathbf{M} \in \mathbb{R}^{m \times n}$ by $\sigma_i(\mathbf{M})$. For $\mathbf{M} \in \mathbb{S}_{\succeq \mathbf{0}}^{d \times d}$, we use $\|\mathbf{M}\|_{\mathrm{op}} := \sigma_1(\mathbf{M})$ to denote the $\ell_2$ operator norm of $\mathbf{M} \in \mathbb{R}^{m \times n}$, and we use $\|\mathbf{M}\|_{\mathrm{F}}$ to denote the Frobenius norm. We use $\mathrm{span}(\{\mathbf{v}_i\}_{i \in [k]})$ to denote the span of a set of vectors, and $\mathrm{range}(\mathbf{A})$, $\mathrm{ker}(\mathbf{A})$, $\mathrm{rank}(\mathbf{A})$ to denote the span of the columns of matrix, the null space and the rank of $\mathbf{A}$. For any subspace $\mathbf{S}$, $\dim(\mathbf{S})$ denotes the dimensionality of the subspace. When $\mathbf{M} \in \mathbb{R}^{m \times n}$ and $S \subseteq [m]$, $T \subseteq [n]$, we use $\mathbf{M}_{S \times T}$ to denote the submatrix indexed by $S, T$.

For a parameter $\sigma > 0$, we say that a distribution $\mathcal{D}$ over $\mathbb{R}^d$ is $\sigma$-sub-Gaussian if for all $\mathbf{u} \in \mathbb{R}^d$,

$$\mathbb{E}_{\mathbf{x} \sim \mathcal{D}} \left[ \exp\left( \mathbf{u}^\top \mathbf{x} \right) \right] \leq \exp \left( \frac{1}{2} \sigma^2 \|\mathbf{u}\|_2^2 \right).$$

If $\sigma = 1$ we say that $\mathcal{D}$ is sub-Gaussian. Any $\sigma$-sub-Gaussian distribution must have mean $\mathbf{0}_d$ (which follows by taking $\mathbf{u} \to \mathbf{0}_d$ and differentiating the logarithm of the moment generating function).

We finally provide notation for procedures often used in the paper. For a vector $\mathbf{v} \in \mathbb{R}^d$ and $k \in [d]$, we use $\mathrm{top}_k(\mathbf{v})$ to denote the vector in $\mathbb{R}^d$ that zeroes out all but the top-$k$ entries of $\mathbf{v}$ by magnitude (breaking ties arbitrarily). For a vector or matrix argument, and a threshold $\tau > 0$, we use $\mathcal{T}_\tau(\cdot)$ to be the vector or matrix that applies the following thresholding operation entrywise:

$$\mathcal{T}_\tau(c) := \begin{cases} c & |c| \geq \tau \\ 0 & \text{else} \end{cases}. \tag{2}$$

Our goal in both Models 1 and 2 is to estimate $\mathbf{v}$ from samples. As is standard, to quantify the performance of our algorithms, we use the sine-squared error:

$$\sin^2 \angle(\mathbf{u}, \mathbf{v}) := 1 - \frac{\langle \mathbf{u}, \mathbf{v} \rangle^2}{\|\mathbf{u}\|_2^2 \|\mathbf{v}\|_2^2}. \tag{3}$$

Typically, our goal will be, for some $\delta, \Delta \in (0,1)^2$, to output $\mathbf{u}$ such that with probability $\geq 1 - \delta$ over the random samples and the algorithm, we have $\sin^2 \angle(\mathbf{u}, \mathbf{v}) \leq \Delta$.

Finally, in Sections 3.1 and 3.2 we consider the natural extension of Model 2 to $k$-PCA, where there are multiple sparse principal components that we wish to detect.

**Model 3** (General sparse $k$-PCA model). *Fix $(s, d, n, k) \in \mathbb{N}^4$ with $(s,k) \in [d]^2$, $\gamma \in (0, \frac{1}{2})$, and $\sigma > 0$. In the* general sparse $k$-PCA model, *there are unknown orthonormal $\{\mathbf{v}_i \in \mathbb{R}^d\}_{i \in [k]}$ that span the top-$k$ eigenspace of an unknown $\mathbf{\Sigma} \in \mathbb{S}_{\succeq \mathbf{0}}^{d \times d}$, satisfying $\frac{\lambda_{k+1}(\mathbf{\Sigma})}{\lambda_k(\mathbf{\Sigma})} \leq 1 - \gamma$, and $|\bigcup_{i \in [k]} \mathrm{supp}(\mathbf{v}_i)| \leq s$. We obtain samples $\{\mathbf{x}_i\}_{i \in [n]} \sim_{\text{i.i.d.}} \mathcal{D}$, a $\sigma$-sub-Gaussian distribution with covariance $\mathbf{\Sigma}$.*

We observe that Model 2 is simply Model 3 specialized to $k = 1$ and $\gamma = 0.1$.

## 3. Main Results

We provide our main sparse PCA results under Models 2 and 3 in this section. We state our main result (Theorem 2) in the more general Model 3 for the sparse subspace estimation problem in Section 3.1 and note that it applies to Model 2 directly (by taking $k = 1$). In Section 3.2 we demonstrate a counterexample to the self-reducibility of the sparse $k$-PCA problem (in the form of Model 3) upon recursing with an approximate sparse 1-PCA oracle. For space considerations, several proofs from this section are deferred to Appendix C.

### 3.1. Sparse subspace estimation

For convenience, in this section we follow the notation of Model 3. We denote the collective support of the top-$k$ eigenspace as $S := \bigcup_{j \in [k]} \mathrm{supp}(\mathbf{v}_j)$, and we let $\mathbf{V} \in \mathbb{R}^{d \times k}$ be the horizontal concatenation of the $\{\mathbf{v}_i\}_{i \in [k]}$, and we denote the horizontal concatenation of a basis of their orthogonal complement subspace $\{\mathbf{v}_i\}_{i \in [k+1, d]}$ as $\mathbf{V}_\perp \in \mathbb{R}^{d \times d - k}$. Before we proceed, we require the following useful result.

**Lemma 1.** *Under Model 3, for every $0 < \beta \leq \gamma$ there exists $p \in [k]$ such that the following hold.*

1. $\dfrac{\lambda_p(\mathbf{\Sigma})}{\lambda_1(\mathbf{\Sigma})} \geq 1 - \beta,$   2. $\dfrac{\lambda_{p+1}(\mathbf{\Sigma})}{\lambda_p(\mathbf{\Sigma})} \leq 1 - \dfrac{\beta}{k}.$

*Proof.* Let $p := \min\{i \in [k] \mid \frac{\lambda_{i+1}(\mathbf{\Sigma})}{\lambda_i(\mathbf{\Sigma})} \leq 1 - \frac{\beta}{k}\}$. Such an $i \in [k]$ always exists since $i = k$ satisfies this bound. We claim that $\frac{\lambda_p(\mathbf{\Sigma})}{\lambda_1(\mathbf{\Sigma})} \geq 1 - \beta$. Indeed since $p$ is the first index witnessing an eigengap,

$$\frac{\lambda_p(\mathbf{\Sigma})}{\lambda_1(\mathbf{\Sigma})} = \prod_{i \in [p-1]} \frac{\lambda_{i+1}(\mathbf{\Sigma})}{\lambda_i(\mathbf{\Sigma})} \geq \left( 1 - \frac{\beta}{k} \right)^p \geq 1 - \beta.$$

$\square$

Using Lemma 1, we denote by $p \in [k]$, the index satisfying

$$\frac{\lambda_p(\mathbf{\Sigma})}{\lambda_1(\mathbf{\Sigma})} \geq 1 - \beta, \quad \frac{\lambda_{p+1}(\mathbf{\Sigma})}{\lambda_p(\mathbf{\Sigma})} \leq 1 - \frac{\beta}{k}, \tag{4}$$

for a choice of $\beta \in (0, \gamma]$ to be specified later. Let $\mathbf{V}_p \in \mathbb{R}^{d \times p}$ be the top-$p$ eigenspace of $\mathbf{\Sigma}$ and $\mathbf{V}_{p, \perp}$ denote the residual orthonormal eigenspace, so that

$$\mathbf{\Sigma} = \mathbf{V}_p \mathbf{\Lambda}_p \mathbf{V}_p^\top + \mathbf{V}_{p, \perp} \mathbf{\Lambda}_{p, \perp} \mathbf{V}_{p, \perp}^\top.$$

Then, we analyze the potential function

$$\psi_t^{(i)} := \left\| \mathbf{V}_p^\top \mathbf{u}_t^{(i)} \right\|_2, \tag{5}$$

as $\mathbf{u}_t^{(i)}$ evolves according to the *restarted truncated power method* (Algorithm 1).

We first require a helper lemma from (Yuan and Zhang, 2013) on the effect of truncation.

**Lemma 2** (Lemma 4, (Yuan and Zhang, 2013)). *Let* $\mathbf{u}, \mathbf{v} \in \mathbb{R}^d$ *be unit vectors with* $\mathrm{nnz}(\mathbf{v}) \leq s$. *Then for all* $r \in [d]$,

$$|\langle \mathrm{top}_r(\mathbf{u}), \mathbf{v} \rangle| \geq |\langle \mathbf{u}, \mathbf{v} \rangle|$$

$$-\sqrt{\frac{s}{r}} \min\left\{ 1, \left(1 + \sqrt{\frac{s}{r}}\right)\left(1 - \langle \mathbf{u}, \mathbf{v} \rangle^2\right) \right\}.$$

We extend Lemma 2 to analyze the effect of truncation on subspace correlation.

**Lemma 3.** *Let* $\mathbf{R} = (\mathbf{r}_1 \dots \mathbf{r}_k) \in \mathbb{R}^{d \times k}$ *be any orthonormal matrix such that* $\left| \bigcup_{j \in [t]} \mathrm{supp}(\mathbf{r}_j) \right| \leq s$ *and let* $\mathbf{u} \in \mathbb{R}^d$ *be a unit vector. Then for all* $r \in [d]$,

$$\left\| \mathbf{R}^\top \mathrm{top}_r(\mathbf{u}) \right\|_2 \geq \left\| \mathbf{R}^\top \mathbf{u} \right\|_2$$

$$-\sqrt{\frac{s}{r}} \min\left\{ 1, \left(1 + \sqrt{\frac{s}{r}}\right)\left(1 - \left\| \mathbf{R}^\top \mathbf{u} \right\|_2^2\right) \right\}.$$

Next we give a simple bound on the runtime of Algorithm 1.

**Lemma 4.** *Algorithm 1 can be implemented in time* $O(nd^2T)$.

*Proof.* Observe that scalar multiplication cannot change the relative rank of entries by magnitude, so we may perform all of the normalization steps in Line 6 at the end of the algorithm in $O(d^2)$ time without loss of generality. For convenience in this proof, write $\mathbf{w}_t^{(i)}$ to be $\mathbf{u}_t^{(i)}$ had none of the normalization steps in Line 6 taken place, and write

$$\mathbf{Z} := \frac{1}{\sqrt{n}} \begin{pmatrix} \mathbf{x}_1 & \mathbf{x}_2 & \dots & \mathbf{x}_n \end{pmatrix} \in \mathbb{R}^{d \times n},$$

so that $\widehat{\boldsymbol{\Sigma}} = \mathbf{Z}\mathbf{Z}^\top$. Finally, write $\mathbf{W}_t$ to be the $d \times d$ matrix whose $i^{\text{th}}$ column is $\mathbf{w}_t^{(i)}$. With this notation in hand, we have the recursion $\mathbf{W}_0 \leftarrow \mathbf{I}_d$, and

$$\mathbf{W}_t \leftarrow \mathrm{top}_r\left(\mathbf{Z}\mathbf{Z}^\top \mathbf{W}_{t-1}\right),$$

where the operation $\mathrm{top}_r$ is applied columnwise. The matrix-matrix multiplication $\mathbf{Z}^\top \mathbf{W}_{t-1}$ can be computed in $O(ndr)$ time, because it consists of $d$ matrix-vector products with $r$-sparse vectors, and computing $\mathbf{Z}(\mathbf{Z}^\top \mathbf{W}_{t-1})$ takes $O(nd^2)$ time. Finally, the $\mathrm{top}_r$ operation takes $O(d^2)$ time. $\square$

Next, we state our main progress lemma on the progress made in a single iteration of Algorithm 1.

**Lemma 5.** *Let* $S \subseteq F \subseteq [d]$, *and let* $\mathbf{I}_F \in \{0,1\}^{d \times d}$ *denote the diagonal matrix with* $[\mathbf{I}_F]_{ii} := \mathbb{1}(i \in F)$. *Let* $\mathbf{x} \in \mathbb{R}^d$, $\|\mathbf{x}\|_2 = 1$ *have support* $F$ *and let* $\rho := \left\| \mathbf{V}_p^\top \mathbf{x} \right\|_2$. *Define*

$$\mathbf{y} := \frac{\widehat{\boldsymbol{\Sigma}}_F \mathbf{x}}{\left\| \widehat{\boldsymbol{\Sigma}}_F \mathbf{x} \right\|_2}, \quad \boldsymbol{\Sigma}_F := \mathbf{I}_F \boldsymbol{\Sigma} \mathbf{I}_F, \quad \widehat{\boldsymbol{\Sigma}}_F := \mathbf{I}_F \widehat{\boldsymbol{\Sigma}} \mathbf{I}_F.$$

*If*

$$\epsilon_F := \frac{1}{\lambda_p(\boldsymbol{\Sigma})} \left\| \boldsymbol{\Sigma}_F - \widehat{\boldsymbol{\Sigma}}_F \right\|_{\mathrm{op}} \leq \frac{\lambda_p(\boldsymbol{\Sigma}) - \lambda_{p+1}(\boldsymbol{\Sigma})}{8\lambda_p(\boldsymbol{\Sigma})},$$

*then for* $\kappa := \frac{\lambda_{p+1}(\boldsymbol{\Sigma})}{\lambda_p(\boldsymbol{\Sigma})}$, *we may lower bound* $\left\| \mathbf{V}_p^\top \mathbf{y} \right\|_2$ *by*

$$\rho\left(1 + \frac{3(1-\kappa)(1 - (\rho + \frac{2\epsilon_F}{1-\kappa})^2)}{16}\right) - \frac{5\epsilon_F}{1-\kappa}.$$

Lemma 5 provides a lower bound on the growth in our potential function, $\psi_t^{(i)}$ (5). We next derive simpler consequences of Lemma 5. We split our analysis into two phases: the early iterations are analyzed in Lemma 6, and the later iterations in Lemma 7. Notably, in Lemma 7, it will be useful to switch our potential from the correlation to the residual.

**Lemma 6.** *Suppose that under Model 3, we have in Algorithm 1 that for some* $i \in [d]$ *and* $t \in [T]$,

$$\psi_{t-1}^{(i)} \in \left[ \sqrt{\frac{p}{s}}, \frac{1}{\sqrt{2}} \right].$$

*Then, for* $\delta \in (0,1)$, *if we take*

$$r = \Omega\left(\frac{s^2 k^2}{\beta^2}\right), \quad n = \Omega\left(\left(\frac{\sigma^2}{\lambda_p(\boldsymbol{\Sigma})}\right)^2 \cdot \frac{s^3 k^6}{\beta^6} \log\left(\frac{d}{\delta}\right)\right)$$

*for appropriate constants, with probability at least* $1 - \delta$, *we have* $\psi_t^{(i)} \geq (1 + \frac{\beta}{16k})\psi_{t-1}^{(i)}$.

**Lemma 7.** *Suppose that under Model 3, we have in Algorithm 1 that for some* $i \in [d]$ *and* $t \in [T]$,

$$\psi_{t-1}^{(i)} \in \left[ \frac{1}{\sqrt{2}}, \sqrt{1-\Delta} \right].$$

*Then, for* $\delta \in (0,1)$, $\Delta \in (0, \frac{1}{2})$, *if we take*

$$r = \Omega\left(\frac{sk^2}{\Delta^2 \beta^2}\right), \quad n = \Omega\left(\left(\frac{\sigma^2}{\lambda_p(\boldsymbol{\Sigma})}\right)^2 \cdot \frac{s^2 k^6}{\Delta^4 \beta^6} \log\left(\frac{d}{\delta}\right)\right)$$

*for appropriate constants, with probability at least* $1 - \delta$, *we have* $\widetilde{\psi}_t^{(i)} \leq (1 - \frac{\beta}{8k})\widetilde{\psi}_{t-1}^{(i)}$ *for*

$$\widetilde{\psi}_t^{(i)} := 1 - (\psi_t^{(i)})^2.$$

We remark that the sample complexities in Lemmas 6 and 7 follow from standard matrix concentration inequalities, recalled in Appendix A, and are bottlenecked by the sampling error required by Lemma 5.

Finally, we are ready to combine our progress bounds from Lemmas 6 and 7. The following quadratic form guarantee on the output of Algorithm 1 follows by taking enough samples for the conditions of Lemmas 6 and 7 to hold, and then accounting for the error of the selection step in Line 9.

**Proposition 1.** *Let* $\delta \in (0,1)$, $\epsilon \in (0, \frac{1}{2})$, *and under Model 3, assume that*

$$n = \Omega\left(\left(\frac{\sigma^2}{\lambda_k(\mathbf{\Sigma})}\right)^2 \left(\frac{s^3 k^6}{\beta^6} + \frac{s^2 k^6}{\epsilon^4 \beta^6}\right) \log\left(\frac{d}{\delta}\right)\right),$$

$$r = \Omega\left(\frac{s^2 k^2}{\beta^2} + \frac{sk^2}{\beta^2\epsilon^2}\right), \quad T = \Omega\left(\frac{k \log\left(\frac{s}{\epsilon}\right)}{\epsilon}\right)$$

*for appropriate constants. Then, in time $O\left(nd^2 T\right)$, Algorithm 1 returns an $r$-sparse unit vector $\mathbf{u}$ such that with probability at least $1 - \delta$, $\langle \mathbf{u}, \mathbf{\Sigma u}\rangle \geq (1 - \epsilon - \beta)\lambda_1(\mathbf{\Sigma})$.*

Proposition 1 gives a quadratic form guarantee on the selected output in Line 9 (as opposed to a correlation bound, as tracked by our potential (5)), because our potential is not an observable quantity. For convenience, in our final theorem statement, we convert this to a direct correlation guarantee with respect to the leading eigenspace $\mathbf{V}$.

**Theorem 2.** *Let $\delta \in (0,1)$, $\Delta \in (0,1)$, and under Model 3, assume that*

$$n = \Omega\left(\left(\frac{\sigma^2}{\lambda_k(\mathbf{\Sigma})}\right)^2 \left(\frac{s^3 k^6}{\Delta^6 \gamma^6} + \frac{s^2 k^6}{\Delta^{10} \gamma^{10}}\right) \log\left(\frac{d}{\delta}\right)\right),$$

$$r = \Omega\left(\frac{s^2 k^2}{\Delta^2 \gamma^2} + \frac{sk^2}{\Delta^4 \gamma^4}\right), \quad T = \Omega\left(\frac{k \log\left(\frac{s}{\Delta\gamma}\right)}{\Delta\gamma}\right)$$

*for appropriate constants. Then, in time $O\left(nd^2 T\right)$, Algorithm 1 returns an $r$-sparse unit vector $\mathbf{u}$ such that with probability at least $1 - \delta$, $\left\|\mathbf{V}^\top \mathbf{u}\right\|_2^2 \geq 1 - \Delta$.*

*Proof.* Let $\epsilon = \beta = \frac{\Delta\gamma}{2}$, which matches the requirements of Lemma 1 and Proposition 1, so that

$$\langle \mathbf{u}, \mathbf{\Sigma u}\rangle \geq (1 - \Delta\gamma)\lambda_1(\mathbf{\Sigma}), \tag{6}$$

with probability $\geq 1 - \delta$ for the stated ranges of $n, r$. Then for $\alpha := \left\|\mathbf{V}_\perp^\top \mathbf{u}\right\|_2$,

$$\begin{aligned}
\langle \mathbf{u}, \mathbf{\Sigma u}\rangle &= \mathbf{u}^\top(\mathbf{V\Lambda V}^\top + \mathbf{V}_\perp \mathbf{\Lambda}_\perp \mathbf{V}_\perp^\top)\mathbf{u} \\
&\leq \lambda_1(\mathbf{\Sigma})(1 - \alpha^2) + \lambda_{k+1}(\mathbf{\Sigma})\alpha^2 \\
&\leq \lambda_1(\mathbf{\Sigma})(1 - \alpha^2) + \lambda_1(\mathbf{\Sigma})\alpha^2(1 - \gamma) \\
&= \lambda_1(\mathbf{\Sigma}) + \lambda_1(\mathbf{\Sigma})\left(\alpha^2(1 - \gamma) - \alpha^2\right) \tag{7} \\
&= \lambda_1(\mathbf{\Sigma}) - \lambda_1(\mathbf{\Sigma})\alpha^2\gamma. \tag{8}
\end{aligned}$$

From (6) and (8), we conclude the desired $\alpha^2 \leq \Delta$. $\qquad\square$

It is often convenient to convert the output candidate sparse principal component $\mathbf{u}$ into a proper estimate, by truncating it to be exactly $s$-sparse. We give a helper result showing that this sparse projection step only worsens the final bound by a constant factor.

**Lemma 8.** *In the setting of Model 3, let $\mathbf{u} \in \mathbb{R}^d$ be an $r$-sparse unit vector with $r \geq s$ satisfying $\left\|\mathbf{V}^\top \mathbf{u}\right\|_2^2 \geq 1 - \Delta$ for $\Delta \in (0, \frac{1}{2})$. Then,*

$$\left\|\mathbf{V}^\top \frac{\mathrm{top}_s(\mathbf{u})}{\|\mathrm{top}_s(\mathbf{u})\|_2}\right\|_2^2 \geq 1 - 5\Delta.$$

*Proof.* Let $\alpha := \|\mathbf{V}^\top\mathbf{u}\|_2 \in [0,1]$, so $\alpha^2 \geq 1 - \Delta$. Applying Lemma 3 to the unit vector $\mathbf{u}$:

$$\begin{aligned}
\|\mathbf{V}^\top\mathrm{top}_s(\mathbf{u})\|_2 &\geq \|\mathbf{V}^\top\mathbf{u}\|_2 - \min\left\{1, 2\left(1 - \|\mathbf{V}^\top\mathbf{u}\|_2^2\right)\right\} \\
&= \alpha - \min\left\{1, 2(1 - \alpha^2)\right\}.
\end{aligned}$$

Since $\Delta \in (0, \frac{1}{2})$ and $1 - \alpha^2 \leq \Delta$, we have $2(1 - \alpha^2) \leq 2\Delta < 1$, hence

$$\|\mathbf{V}^\top\mathrm{top}_s(\mathbf{u})\|_2 \geq \alpha - 2(1 - \alpha^2) \geq \alpha - 2\Delta \geq \sqrt{1 - 5\Delta}.$$

The result follows from $\|\mathrm{top}_s(\mathbf{u})\|_2 \leq 1$. $\qquad\square$

### 3.2. Barrier for sparse PCA deflation methods

To solve the $k$-sparse PCA problem in Model 3, a natural strategy is to use a *deflation method* that repeatedly projects out learned sparse components, and recursively calls a 1-sparse PCA oracle on the remaining "deflated" matrix. We give pseudocode for such a method in Algorithm 2.

---

**Algorithm 2:** $\mathsf{kSPCA}(\mathbf{\Sigma}, k, h, \mathcal{O}_{1\mathrm{SPCA}})$

---

1 **Input:** $\mathbf{\Sigma} \in \mathbb{S}_{\succeq 0}^{d \times d}$, eigenspace dimension $k \in \mathbb{N}$, sparsity parameter $h$, 1-SPCA oracle $\mathcal{O}_{1\mathrm{SPCA}}$
2 $\mathbf{P}_0 \leftarrow \mathbf{I}_d$
3 **for** $i \in [k]$ **do**
4 $\quad \mathbf{M}_i \leftarrow \mathbf{P}_{i-1}\mathbf{\Sigma}\mathbf{P}_{i-1}$
5 $\quad \mathbf{u}_i \leftarrow \mathcal{O}_{1\mathrm{SPCA}}\left(\mathbf{M}_i, \mathbf{P}_{i-1}\right)$
6 $\quad \mathbf{P}_i \leftarrow \mathbf{P}_{i-1} - \mathbf{u}_i\mathbf{u}_i^\top$
7 **end**
8 **return** $\mathbf{U} \leftarrow \{\mathbf{u}_i\}_{i \in [k]} \in \mathbb{R}^{d \times k}$

---

Such a strategy was popularized by (Mackey, 2008) (e.g., Algorithm 1 of the paper is essentially Algorithm 2), and its approximation tolerance was analyzed rigorously by (Jambulapati et al., 2024) in the dense setting ($s = d$). While we do not fully specify the guarantees of the sparse 1-PCA oracle on Line 5, at minimum, the following requirements are natural for the recursion to make sense.

1. $\mathbf{u}_i$ is a unit vector in $\mathrm{range}(\mathbf{P}_{i-1})$.

2. $\mathbf{u}_i$ correlates with the top eigenspace of $\mathbf{P}_{i-1}\mathbf{\Sigma}\mathbf{P}_{i-1}$ (as it solves a residual 1-PCA problem).

3. $\mathbf{u}_i$ should itself be sparse.

Under these properties, one may hope that the residual problem that $\mathbf{u}_i$ is solving is itself a sparse PCA problem in the sense of Model 3, so we can apply an oracle with the guarantees of Theorem 2 and iterate. This would require the top eigenspace of the deflated matrix $\mathbf{P}_{i-1}\mathbf{\Sigma}\mathbf{P}_{i-1}$ to itself be in the form required by Model 3, i.e., it should be sparsely supported. Because the original top eigenspace of $\mathbf{\Sigma}$ is sparse, and we only project out sparse components that are well-aligned with the top eigenspace of $\mathbf{\Sigma}$, it is plausible that this sparsity property persists.

We demonstrate a major barrier to this strategy, by showing that even when $s = 2$ in Model 3, and the gap parameter $\gamma$ and correlation with the top eigenvector $\Delta$ are allowed to vary arbitrarily, the top eigenspace of the residual matrix can be made arbitrarily dense in one iteration.

**Lemma 9.** *Let* $(\Delta, \gamma) \in (0,1)^2$ *and* $d \geq 4$. *There exists* $\mathbf{\Sigma} \in \mathbb{S}_{\succeq \mathbf{0}}^{d \times d}$ *such that* $(1-\gamma)\lambda_2(\mathbf{\Sigma}) \geq \lambda_3(\mathbf{\Sigma})$, *and* $\mathbf{v}_1, \mathbf{v}_2$, *the top two eigenvectors of* $\mathbf{\Sigma}$, *are both supported on a* 2-*sparse set, yet for a* 3-*sparse unit* $\mathbf{u} \in \mathbb{R}^d$ *with* $\langle \mathbf{u}, \mathbf{v}_1 \rangle^2 \geq 1 - \Delta$, *if* $\mathbf{P} := \mathbf{I}_d - \mathbf{u}\mathbf{u}^\top$, *then* $|\text{supp}(\mathbf{v}_1(\mathbf{P}\mathbf{\Sigma}\mathbf{P}))| = d$.

Lemma 9 is a strong barrier towards provable guarantees for strategies such as Algorithm 2, as it holds even if the correlation $\Delta$ and gap $\gamma$ are arbitrarily small and depend on each other. Moreover, it holds even in the first iteration, and when the sparse 1-PCA oracle has strong correlation to the top eigenvector (not just the sparse eigenspace $\text{span}(\{\mathbf{v}_1, \mathbf{v}_2\})$). We leave it as an important open problem to design a provable combinatorial method for solving sparse $k$-PCA (e.g., under Model 3) in full generality, whether via a modified deflation method or otherwise.

## 4. Experiments

This section presents a series of empirical tests over both synthetic and real-world datasets to compare the efficiency and performance of our new method with heuristic and SDP-based methods. For space considerations, we defer additional experiments (an ablation test to quantify our method's dependence on various parameters in practice, as well as an evaluation on a real-world dataset) to Appendix F.

**Runtime-accuracy tradeoff.** We first empirically compare the runtime efficiency of our proposed algorithm (RTPM, Algorithm 1) with the SDP-based method and other heuristic combinatorial algorithms discussed in Appendix B. For the SDP-based approach, we utilize the Fantope Projection and Selection (FPS) method from (Vu et al., 2013), which implements an alternating direction method of multipliers (ADMM)-based algorithm, although without provable convergence guarantees. State-of-the-art SDP solvers with provable convergence guarantees take time $\Omega(d^{2\omega})$ in theory (Huang et al., 2022), and practical SDP solvers with provable guarantees have even larger runtimes, making them

infeasible even for moderate dimensionalities, as the sparse PCA SDP has $\Theta(d^2)$ constraints.

The combinatorial algorithms we compare with are DiagThresh (Algorithm 3), CovThresh (Algorithm 4) and GreedyCorr (Algorithm 5). The details are found in Appendix B. Some methods we consider involve tunable parameters, such as the number of iterations, which naturally trades off runtime and performance. Other methods are one-shot and not tunable. For a fair comparison, we report the trajectory of the squared cosine correlation versus cumulative runtime for tunable methods. Non-tunable methods appear as scattered points under the same configuration.

We generate $n$ samples from Model 1 and the counterexample in Lemma 13, and test the runtime and accuracy for all the methods above. The latter experiment is to illustrate runtime behavior on a non-spiked sparse PCA model as defined in Model 2. For a meaningful comparison, we choose parameters such that the targeted heuristics do not fail.

As seen in Figure 1, our RTPM is the only algorithm that consistently achieves high correlation with the target vector across all sample sizes and covariance instances.

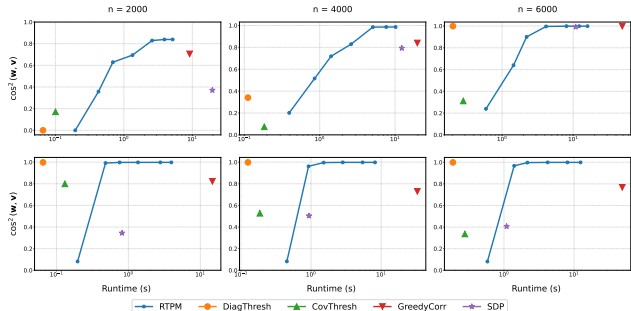

*Figure 1.* Runtime versus accuracy for RTPM, heuristic and SDP-based methods. First row: spiked identity in Model 1 with $d = 1000$ and $s = 8$. Second row: counterexample against GreedyCorr presented in Lemma 13 with $d = 1000$, $s = 8$, and $\lambda_1(\mathbf{\Sigma}) = 1.2$, $\lambda_2(\mathbf{\Sigma}) = 0.8$. RTPM runs with $r = s$ and the relaxation coefficient for the SDP-based method is set as suggested in (Vu et al., 2013). The runtime is an average of 10 runs.

**Counterexamples.** For each counterexample discussed in Appendix B, we will compare the performance of the targeted heuristic method, RTPM, and the SDP-based method. Some parameters in data generation and algorithmic implementations are not set exactly as in their theoretical analysis, as they often rely on some universal constants that are impractical to compute. Instead, we adopt a pragmatic strategy to combine heuristic choices with a coarse grid search.

The experiments in Figure 2 confirm the theoretical guaranties that our counterexamples successfully fool the simple heuristic methods. Moreover, these tests also indicate that our method is able to overcome these counterexamples and performs well on many instances under the general Model 2.

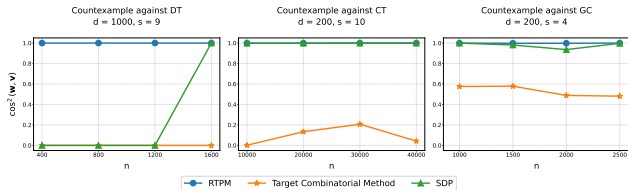

*Figure 2.* Performance on counterexamples. In each subplot, we vary the sample size $n$ under fixed $d, k$, and compare the output correlation achieved RTPM, the targeted heuristics and the SDP-based method. The dataset parameter except $(n, d, s)$ are set as following: the left plot uses $\lambda_1 = 1.0$, $\lambda_2 = 0.5$, $\lambda_2/\lambda_3 = 2.1$ and $\lambda_2/\lambda_4 = 2.2$; the middle plot uses $u = 25$, $r = 6$, $\theta = 1$ and $c = 0.25$; the right plot uses parameters in Lemma 13. The RTPM method is run for 40 iterations and the relaxation coefficient for SDP-based method is set as suggested in (Vu et al., 2013).

## Acknowledgments

We thank Eric Price for several helpful conversations at an earlier stage of this project, which led to the counterexample in Section B.2. KT thanks Ankit Pensia for several early conversations on the topic of sparse PCA, which motivated this work. SK gratefully acknowledges funding support from the Amazon AI PhD Fellowship. PS gratefully acknowledges NSF grants 2217069 and CCF-2505865.

## Impact Statement

This paper presents work whose goal is to advance the field of Machine Learning. There are many potential societal consequences of our work, none which we feel must be specifically highlighted here.

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

# A. Additional preliminaries

We require the following standard facts to bound the approximation error of random sampling, under our Models 1, 2, and 3.

**Fact 1** (Lemma B.3, (Zhu and Li, 2016)). *For some $\mu > 0$ and $\tau > 0$, let orthonormal $\mathbf{U}$ have columns spanning the eigenspace of $\mathbf{A} \in \mathbb{S}_{\succeq \mathbf{0}}^{d \times d}$ corresponding to eigenvalues $\leq \mu$, and let orthonormal $\mathbf{V}$ have columns spanning the eigenspace of $\mathbf{B} \in \mathbb{S}_{\succeq \mathbf{0}}^{d \times d}$ corresponding to eigenvalues $\geq \mu + \tau$. Then,*

$$\left\| \mathbf{U}^\top \mathbf{V} \right\|_{\mathrm{op}} \leq \frac{\|\mathbf{A} - \mathbf{B}\|_{\mathrm{op}}}{\tau}.$$

**Fact 2** (Lemma 6.26, (Wainwright, 2019)). *Let $\mathcal{D}$ be a $\sigma$-sub-Gaussian distribution with covariance $\mathbf{\Sigma} \in \mathbb{S}_{\succeq \mathbf{0}}^{d \times d}$, let $\{\mathbf{x}_i\}_{i \in [n]} \sim_{\text{i.i.d.}} \mathcal{D}$, and let $\widehat{\mathbf{\Sigma}} := \frac{1}{n} \sum_{i \in [n]} \mathbf{x}_i \mathbf{x}_i^\top$. There exists a universal constant $C > 0$ such that for all $\delta \in (0, 1)$,*

$$\Pr\left[ \max_{(i,j) \in [d] \times [d]} \left| \widehat{\mathbf{\Sigma}}_{ij} - \mathbf{\Sigma}_{ij} \right| \geq C\sigma^2 \left( \sqrt{\frac{\log(\frac{d}{\delta})}{n}} + \frac{\log(\frac{d}{\delta})}{n} \right) \right] \leq \delta.$$

**Fact 3** (Exercise 4.7.3, (Vershynin, 2018)). *Let $\mathcal{D}$ be a $\sigma$-sub-Gaussian distribution with covariance $\mathbf{\Sigma} \in \mathbb{S}_{\succeq \mathbf{0}}^{d \times d}$, let $\{\mathbf{x}_i\}_{i \in [n]} \sim_{\text{i.i.d.}} \mathcal{D}$, and let $\widehat{\mathbf{\Sigma}} := \frac{1}{n} \sum_{i \in [n]} \mathbf{x}_i \mathbf{x}_i^\top$. There exists a universal constant $C > 0$ such that for all $\delta \in (0, 1)$,*

$$\Pr\left[ \left\| \widehat{\mathbf{\Sigma}} - \mathbf{\Sigma} \right\|_{\mathrm{op}} \geq C\sigma^2 \left( \sqrt{\frac{d + \log(\frac{1}{\delta})}{n}} + \frac{d + \log(\frac{1}{\delta})}{n} \right) \|\mathbf{\Sigma}\|_{\mathrm{op}} \right] \leq \delta.$$

As a simple corollary of Fact 3, we derive a concentration bound for all submatrices.

**Corollary 1.** *In the setting of Fact 3, let $s \in [d]$. There exists a universal constant $C > 0$ such that for all $\delta \in (0, 1)$,*

$$\Pr\left[ \max_{\substack{S \subseteq [d] \\ |S| \leq s}} \left\| \widehat{\mathbf{\Sigma}}_{S \times S} - \mathbf{\Sigma}_{S \times S} \right\|_{\mathrm{op}} \geq C\sigma^2 \left( \sqrt{\frac{s \log(d) + \log(\frac{1}{\delta})}{n}} + \frac{s \log(d) + \log(\frac{1}{\delta})}{n} \right) \right] \leq \delta.$$

*Proof.* Note that there are $\binom{d}{s} \leq d^s$ such submatrices. It suffices to apply Fact 3 to each such submatrix with failure probability $\delta \leftarrow \frac{\delta}{d^s}$ and dimension $d \leftarrow s$, and take a union bound. $\square$

# B. Counterexamples

In this section, we consider existing combinatorial algorithms with runtime $d^2 \cdot \mathrm{poly}(s, \log(d))$ for solving sparse PCA under Model 1, and exhibit explicit counterexamples against their success under Model 2. All of our counterexamples take $\gamma = \Theta(1)$ and $\sigma = 1$ in Model 2.

## B.1. Diagonal thresholding

We first discuss the diagonal thresholding algorithm initially proposed in (Amini and Wainwright, 2008; Johnstone and Lu, 2009b). The algorithm simply selects the top-$s$ coordinates of the diagonal of the sample covariance matrix as an estimate for the support of the principal component $\mathbf{v}$ (Algorithm 3), which is known to succeed in recovering heavy coordinates of $\mathbf{v}$ under Model 1 (see e.g., Proposition 1, (Amini and Wainwright, 2008)).

We restate a recent counterexample by (Kumar and Sarkar, 2024) which shows that under Model 2, diagonal thresholding can fail to detect any of the support elements.

**Lemma 10** (Proposition 3.4, (Kumar and Sarkar, 2024)). *Under Model 2, for any $n \geq C_1 s^2 \log d$ where $C_1 > 0$ is a universal constant, there is a matrix $\mathbf{\Sigma}$ that satisfies $\gamma = \Theta(1)$, but Algorithm 3 applied to any 1-sub-Gaussian $\mathcal{D}$ with covariance $\mathbf{\Sigma}$ outputs $\mathbf{u}$ satisfying $\sin^2 \angle(\mathbf{u}, \mathbf{v}) = 1$ with probability $\geq \frac{1}{2}$.*

We remark that requiring $n = \Omega(s^2 \log d)$ is unrestrictive, as this sample complexity is needed in all known algorithms for sparse PCA (and is conjectured to be necessary for any polynomial-time algorithms to exist, even under the simpler Model 1 (Berthet and Rigollet, 2013)).

---

**Algorithm 3:** DiagThresh($\{\mathbf{x}_i\}_{i\in[n]}, s$)

---

1 **Input:** Dataset $\{\mathbf{x}_i \in \mathbb{R}^d\}_{i\in[n]}$, sparsity hyperparameter $s \in [d]$

2 $\widehat{\boldsymbol{\Sigma}} \leftarrow \frac{1}{n} \sum_{i\in[n]} \mathbf{x}_i \mathbf{x}_i^\top$

3 $S \leftarrow s$ largest indices $i \in [d]$ ordered by $\widehat{\boldsymbol{\Sigma}}_{ii}$

4 **return** $\mathbf{u} \leftarrow \mathbf{v}_1(\widehat{\boldsymbol{\Sigma}}_{S\times S})$

---

## B.2. Covariance thresholding

We next discuss a counterexample for a covariance thresholding algorithm (reproduced as Algorithm 4) that was shown to solve sparse PCA under Model 1 in (Deshpande and Montanari, 2016), and more generally analyzed for sparse covariance estimation in Theorem 6.23, (Wainwright, 2019). The algorithm picks a parameter $\tau > 0$, thresholds the entries of the sample covariance matrix at $\tau$, and computes the top eigenvector of this thresholded matrix. This algorithm is more generally known to succeed with an $\Omega(s^2 \log d)$ sample complexity under a row-wise and column-wise $s$-sparsity assumption on the covariance.

---

**Algorithm 4:** CovThresh($\{\mathbf{x}_i\}_{i\in[n]}, s, \tau$)

---

1 **Input:** Dataset $\{\mathbf{x}_i \in \mathbb{R}^d\}_{i\in[n]}$, sparsity hyperparameter $s \in [d]$, threshold $\tau > 0$

2 $\widehat{\boldsymbol{\Sigma}} \leftarrow \frac{1}{n} \sum_{i\in[n]} \mathbf{x}_i \mathbf{x}_i^\top$

3 **return** $\mathbf{u} \leftarrow \mathbf{v}_1(\mathcal{T}_\tau(\widehat{\boldsymbol{\Sigma}}))$      `// See (2) for the definition of ` $\mathcal{T}_\tau$`.`

---

We require one helper fact for our counterexample.

**Lemma 11.** *Let* $\mathbf{M} \in \mathbb{R}^{d\times d}$ *have* $|\mathbf{M}_{ij}| \leq \tau$ *for all* $(i,j) \in [d] \times [d]$. *Then,* $\|\mathbf{M}\|_{\mathrm{op}} \leq d\tau$.

*Proof.* Take $\tau = 1$ without loss of generality by scaling. For any unit vectors $\mathbf{u}, \mathbf{v}$ we have

$$\mathbf{u}^\top \mathbf{M} \mathbf{v} = \sum_{(i,j)\in[d]\times[d]} \mathbf{u}_i \mathbf{v}_j \mathbf{M}_{ij} \leq \sqrt{\sum_{(i,j)\in[d]\times[d]} \mathbf{u}_i^2 \mathbf{M}_{ij}} \sqrt{\sum_{(i,j)\in[d]\times[d]} \mathbf{v}_j^2 \mathbf{M}_{ij}} \leq d.$$

$\square$

**Lemma 12.** *Under Model 2, for any* $n \geq C_1 s^2 \log d$ *and* $\tau \geq C_2 \sqrt{\frac{\log d}{n}}$ *where* $C_1, C_2 > 0$ *are universal constants, there is a matrix* $\boldsymbol{\Sigma}$ *that satisfies* $\gamma = \Theta(1)$, *but Algorithm 4 applied to any* 1-*sub-Gaussian* $\mathcal{D}$ *with covariance* $\boldsymbol{\Sigma}$ *outputs* $\mathbf{u}$ *satisfying* $\sin^2 \angle(\mathbf{u}, \mathbf{v}) = 1$ *with probability* $\geq \frac{1}{2}$.

*Proof.* Throughout the proof, fix some $u \in \mathbb{N}$ satisfying

$$4s + \frac{8}{\tau} \leq u \leq \frac{1}{144\tau^2}.$$

Let $r := \frac{u-1}{4}$ and $p := \frac{r}{u-1} = \frac{1}{4}$. There exists an $r$-regular graph $G$ (see (Friedman, 2003)) on $u$ vertices with adjacency matrix $\mathbf{A} \in \{0,1\}^{u\times u}$, and

$$\max_{i\geq 2} |\lambda_i(\mathbf{A})| \leq 3\sqrt{r}, \quad \lambda_1(\mathbf{A}) = r, \quad \mathbf{v}_1(\mathbf{A}) := \frac{1}{\sqrt{u}} \mathbf{1}_u.$$

At a high level, our strategy is to construct $\boldsymbol{\Sigma}$ involving $\mathbf{A}$ such that without thresholding, only the second eigenvalue of $\mathbf{A}$ is exposed, but such that thresholding at $\tau$ exposes the top eigenvalue and changes the top eigenvector of $\boldsymbol{\Sigma}$, even after accounting for sampling error. Define the matrix

$$\mathbf{H} := \mathbf{A} - p\left(\mathbf{1}_u \mathbf{1}_u^\top - \mathbf{I}_u\right).$$

Then $\mathbf{H}\mathbf{1}_u = \mathbf{0}_u$. Moreover, for all vectors $\mathbf{u}$ such that $\mathbf{u}^\top \mathbf{1}_u = 0$, $\mathbf{H}\mathbf{u} = (\mathbf{A} + p\mathbf{I}_u)\,\mathbf{u}$. Thus,

$$\|\mathbf{H}\|_{\mathrm{op}} \le \max\left\{ p, \max_{i \ge 2} |\lambda_i(\mathbf{A})| + p \right\} \le 3\sqrt{r} + p \le 2\sqrt{u}. \tag{9}$$

Partition $[d] := S \cup U$ where $S := [s]$, $U \in [d] \setminus S$. Let $\mathbf{v} := \frac{1}{\sqrt{s}}\mathbf{1}_s$, and define $\mathbf{\Sigma}_{S \times S} := \frac{1}{2}(\mathbf{I}_k + \mathbf{v}\mathbf{v}^\top)$, $\mathbf{\Sigma}_{U \times U} := \frac{1}{2}(\mathbf{I}_u + 3\tau\mathbf{H})$, and let all other entries of $\mathbf{\Sigma}$ be 0. Note that for $i \ne j \in U$,

$$\mathbf{\Sigma}_{ij} = \begin{cases} \frac{3}{2}\tau(1-p) = \frac{9\tau}{8} & \mathbf{A}_{ij} = 1 \\ -\frac{3}{2}\tau p = -\frac{3\tau}{8} & \mathbf{A}_{ij} = 0 \end{cases}. \tag{10}$$

Also, by using (9), $\|\mathbf{\Sigma}_{U \times U} - \frac{1}{2}\mathbf{I}_u\|_{\mathrm{op}} \le 3\tau\sqrt{u} \le \frac{1}{4}$, so by Weyl's perturbation theorem, for all $i \in [u]$,

$$\frac{1}{4} \le \lambda_i(\mathbf{\Sigma}_{U \times U}) \le \frac{3}{4}.$$

Therefore the top eigenvalue of $\mathbf{\Sigma}$ is achieved by the $S \times S$ block, and $\frac{\lambda_1(\mathbf{\Sigma})}{\lambda_2(\mathbf{\Sigma})} \ge \frac{4}{3}$.

Next, following notation from Algorithm 4, with probability $\ge \frac{1}{2}$,

$$\max_{(i,j) \in [d] \times [d]} \left| \widehat{\mathbf{\Sigma}}_{ij} - \mathbf{\Sigma}_{ij} \right| \le \frac{\tau}{8}, \quad \left\| \widehat{\mathbf{\Sigma}}_{S \times S} - \mathbf{\Sigma}_{S \times S} \right\|_{\mathrm{op}} \le \frac{1}{2}, \tag{11}$$

so long as $C_1$ and $C_2$ in the statement are large enough functions of the constant $C$ in Corollary 1 and Fact 2. Condition on (11) henceforth. Then, if $\mathbf{\Sigma}_{ij} = 0$, $\mathcal{T}_\tau(\widehat{\mathbf{\Sigma}}_{ij}) = 0$, and using (10), for $i \ne j \in U$,

$$\widehat{\mathbf{\Sigma}}_{ij} \begin{cases} \ge \tau & \mathbf{A}_{ij} = 1 \\ \in [-\frac{\tau}{2}, -\frac{\tau}{4}] & \mathbf{A}_{ij} = 0 \end{cases}.$$

Now, $\mathcal{T}_\tau(\widehat{\mathbf{\Sigma}})_{U \times U}$ is entrywise nonnegative and dominates $\frac{1}{4}\mathbf{I}_u + \tau\mathbf{A}$ entrywise. By the Perron-Frobenius theorem, if $\mathbf{X}$ and $\mathbf{Y}$ are symmetric and entrywise nonnegative matrices such that $\mathbf{X}_{ij} \ge \mathbf{Y}_{ij}$ for all indices $(i,j)$, then $\|\mathbf{X}\|_{\mathrm{op}} \ge \|\mathbf{Y}\|_{\mathrm{op}}$. Consequently,

$$\left\| \mathcal{T}_\tau(\widehat{\mathbf{\Sigma}})_{U \times U} \right\|_{\mathrm{op}} \ge \frac{1}{4} + \tau r \ge \frac{\tau u}{4} \ge 2 + s\tau.$$

However, by using (11) and Lemma 11,

$$\left\| \mathcal{T}_\tau(\widehat{\mathbf{\Sigma}})_{S \times S} \right\|_{\mathrm{op}} \le \lambda_1(\mathbf{\Sigma}_{S \times S}) + \left\| \widehat{\mathbf{\Sigma}}_{S \times S} - \mathbf{\Sigma}_{S \times S} \right\|_{\mathrm{op}} + \left\| \mathcal{T}_\tau(\widehat{\mathbf{\Sigma}})_{S \times S} - \widehat{\mathbf{\Sigma}}_{S \times S} \right\|_{\mathrm{op}} \le \frac{3}{2} + s\tau.$$

Therefore, if (11) holds, then for sufficiently large $s$, the leading eigenvector, $\mathbf{v}_1(\mathcal{T}_\tau(\widehat{\mathbf{\Sigma}}))$, output by Algorithm 4 lies completely in the $U \times U$ block, orthogonal to $\mathbf{v}$. $\qquad\square$

To remark briefly on the parameter restrictions in Lemma 12, the lower bound on $n$ is again essentially without loss of generality, and the lower bound on $\tau$ is the standard setting of the entrywise threshold parameter suggested in (Wainwright, 2019) (up to a constant factor). Our counterexample is quite robust within these ranges, e.g., if the sample size $n$ is taken sufficiently large, Lemma 12 shows that no setting of the threshold $\tau$ can ever recover $\mathbf{v}$ to nontrivial error.

### B.3. Greedy correlation

Finally, we consider a natural analog of an algorithm proposed in recent work (Błasiok et al., 2024), which studied a semi-random variant of the planted clique problem, introduced earlier in (Charikar et al., 2017). The (Błasiok et al., 2024) algorithm proceeded via a combinatorial "greedy correlation" heuristic that aims to find many candidate sets of correlated vertices, one of which overlaps well with the true planted clique. Due to the intimate connection between the planted clique and sparse PCA problems (Berthet and Rigollet, 2013), it is natural to conjecture that a similar combinatorial heuristic robustly solves Model 2.

In Algorithm 5, we reproduce this heuristic for sparse PCA, where $\mathbf{M}_{j:}$ denotes the $j^{\text{th}}$ row of a matrix $\mathbf{M}$.[2] Particularly, given a seed index $i^\star \in [d]$, the algorithm greedily computes the rows most correlated with $\widehat{\boldsymbol{\Sigma}}_{i^\star:}$ as a candidate support. We can then brute force over all $i^\star$, if we can ensure that when $i^\star \in \text{supp}(\mathbf{v})$, the most correlated indices to $i^\star$ have nontrivial overlap with $\text{supp}(\mathbf{v})$.

---

**Algorithm 5:** GreedyCorr($\{\mathbf{x}_i\}_{i \in [n]}, s, i^\star$)

---

**1 Input:** Dataset $\{\mathbf{x}_i \in \mathbb{R}^d\}_{i \in [n]}$, sparsity hyperparameter $s \in [d]$, seed coordinate $i^\star \in [d]$

**2** $\widehat{\boldsymbol{\Sigma}} \leftarrow \frac{1}{n} \sum_{i \in [n]} \mathbf{x}_i \mathbf{x}_i^\top$

**3** $G \leftarrow s$ largest indices $i \in [d]$ ordered by $|\langle \widehat{\boldsymbol{\Sigma}}_{i^\star:}, \widehat{\boldsymbol{\Sigma}}_{i:} \rangle|$

**4 return** $\mathbf{u} \leftarrow \mathbf{v}_1(\widehat{\boldsymbol{\Sigma}}_{G \times G})$

---

The following lemma demonstrates that Algorithm 5 can fail to recover more than a single element in the true support under Model 2, even when seeded with an index from the true support.

**Lemma 13.** *Under Model 2, for any $n \geq C_1 s^2 \log d$ where $C_1 > 0$ is a universal constant, there is a matrix $\boldsymbol{\Sigma}$ that satisfies $\gamma = \Theta(1)$, but Algorithm 3 applied to any 1-sub-Gaussian $\mathcal{D}$ with covariance $\boldsymbol{\Sigma}$ outputs $\mathbf{u}$ satisfying $\sin^2 \angle(\mathbf{u}, \mathbf{v}) \geq 1 - \frac{1}{s}$ with probability $\geq \frac{1}{2}$, even when $i^\star \in \text{supp}(\mathbf{v})$.*

*Proof.* Throughout this proof, let $S := [s]$, $d = 2s - 1$, $\mathbf{v} := \frac{1}{\sqrt{s}} \mathbf{1}_s$, and fix $i^\star = 1$. By Lemma 14, there is an orthonormal set $\{\mathbf{g}_r\}_{r \in [s-1]} \cup \{\mathbf{v}\}$ such that $\langle \mathbf{g}_r, \mathbf{e}_1 \rangle = \frac{1}{\sqrt{s}}$ for all $r \in [s - 1]$. Let

$$\mathbf{u}_r := \frac{1}{\sqrt{2}} \mathbf{g}_r + \frac{1}{\sqrt{2}} \mathbf{e}_{s+r-1}, \text{ for all } r \in [s - 1].$$

Then $\{\mathbf{u}_r\}_{r \in [s-1]} \cup \{\mathbf{v}\}$ is an orthonormal set of vectors. We define

$$\boldsymbol{\Sigma} := \mathbf{v}\mathbf{v}^\top + 0.9 \sum_{r \in [s-1]} \mathbf{u}_r \mathbf{u}_r^\top.$$

Then $\boldsymbol{\Sigma}$ has one eigenvector $\mathbf{v}$ with eigenvalue 1, and $s - 1$ eigenvectors (with equal span to $\{\mathbf{u}_r\}_{r \in [s-1]}$) with eigenvalue 0.9, so it meets the conditions of Model 2.

Next, for any $\mathbf{M} \in \mathbb{S}^{d \times d}$ and $(i, j) \in [d] \times [d]$, $\langle \mathbf{M}_{i:}, \mathbf{M}_{j:} \rangle = (\mathbf{M}^2)_{ij}$. Thus Algorithm 5 computes $S$ which is the largest $s$ entries of $\widehat{\boldsymbol{\Sigma}}_{1:}^2$. Our main goal is to establish that with probability $\geq \frac{1}{2}$,

$$\min_{s+1 \leq j \leq 2s-1} \left| (\widehat{\boldsymbol{\Sigma}}^2)_{1j} \right| > \max_{2 \leq j \leq s} \left| (\widehat{\boldsymbol{\Sigma}}^2)_{1j} \right|. \tag{12}$$

If (12) occurs, then all coordinates $s + 1 \leq j \leq 2s - 1$ will necessarily be included in $G$ computed by Algorithm 5, and hence it can only pick up at most one true entry of $\mathbf{v}$, proving the claim.

We now prove that (12) occurs under a high-probability event. Specifically, for the rest of the proof, let $\mathbf{E} := \widehat{\boldsymbol{\Sigma}} - \boldsymbol{\Sigma}$ be the sampling error, and condition on the event

$$\max_{(i,j) \in [d] \times [d]} |\mathbf{E}_{ij}| \leq \frac{1}{20s}, \tag{13}$$

which occurs with probability $\geq \frac{1}{2}$ by Fact 2. Moreover,

$$\boldsymbol{\Sigma}^2 = \mathbf{v}\mathbf{v}^\top + 0.81 \sum_{r \in [s-1]} \mathbf{u}_r \mathbf{u}_r^\top,$$

---

[2] The strongest result in (Błasiok et al., 2024) actually applies the greedy correlation heuristic to a tensored adjacency matrix. However, even the untensored variant gives semi-random planted clique recovery for sublinear clique sizes $k \approx n^{\frac{3}{4}}$ (cf. Section 2.1, (Błasiok et al., 2024)), so it is natural to conjecture that Algorithm 5 obtains nontrivial recovery at a (possibly lossy) poly($s, \log(d)$) sample complexity, for the semi-random variant of sparse PCA in Model 2.

so that for $2 \leq j \leq s$,

$$\left|(\mathbf{\Sigma}^2)_{1j}\right| = \left|\mathbf{e}_1^\top \left(\mathbf{v}\mathbf{v}^\top + 0.81 \sum_{r \in [s-1]} \mathbf{u}_r \mathbf{u}_r^\top\right) \mathbf{e}_j\right| = \left|\frac{1}{s} + \frac{0.81}{2} \cdot \mathbf{e}_1^\top \left(\mathbf{I}_s - \mathbf{v}\mathbf{v}^\top\right) \mathbf{e}_j\right| \leq \frac{1}{s}. \tag{14}$$

In contrast, for $s + 1 \leq j \leq 2s - 1$, the only $\mathbf{u}_r$ not orthogonal to $\mathbf{e}_j$ satisfies $s + r - 1 = j$, so

$$\left|(\mathbf{\Sigma}^2)_{1j}\right| = \left|\mathbf{e}_1^\top \left(\mathbf{v}\mathbf{v}^\top + 0.81 \sum_{r \in [s-1]} \mathbf{u}_r \mathbf{u}_r^\top\right) \mathbf{e}_j\right| = 0.81 \left|\mathbf{e}_1^\top \mathbf{u}_{j+1-s} \mathbf{u}_{j+1-s}^\top \mathbf{e}_j\right| \geq \frac{0.4}{\sqrt{s}}. \tag{15}$$

Thus (12) holds at the population level. We now consider the effect of sampling. For any $j \in [d]$,

$$\left|(\widehat{\mathbf{\Sigma}}^2)_{1j} - (\mathbf{\Sigma}^2)_{1j}\right| = \left|\sum_{\ell \in [d]} \mathbf{\Sigma}_{1\ell} \mathbf{E}_{\ell j} + \mathbf{E}_{1\ell} \mathbf{\Sigma}_{\ell j} + \mathbf{E}_{1\ell} \mathbf{E}_{\ell j}\right|$$

$$\leq \frac{1}{20s} \left(\|\mathbf{\Sigma}_{1:}\|_1 + \|\mathbf{\Sigma}_{j:}\|_1\right) + d\left(\frac{1}{20s}\right)^2 \leq \frac{1}{20s}\left(\|\mathbf{\Sigma}_{1:}\|_1 + \|\mathbf{\Sigma}_{j:}\|_1\right) + \frac{1}{200s},$$

where we used the assumption (13). Now for any coordinate $i \in [d]$, we must have $\|\mathbf{\Sigma}_{i:}\|_2 \leq 1$ (as $\mathbf{\Sigma}_{i:} = \mathbf{\Sigma}\mathbf{e}_i$ and $\|\mathbf{\Sigma}\|_{\mathrm{op}} \leq 1$), so $\|\mathbf{\Sigma}_{i:}\|_1 \leq \sqrt{d}$. In conclusion,

$$\left|(\widehat{\mathbf{\Sigma}}^2)_{1j} - (\mathbf{\Sigma}^2)_{1j}\right| \leq \frac{\sqrt{d}}{10s} + \frac{1}{200s} \leq \frac{0.15}{\sqrt{s}},$$

so the separation between (14) and (15) persists under sampling error and (12) holds true.

$\square$

## C. Deferred Proofs from Section 3

**Lemma 3.** *Let* $\mathbf{R} = (\mathbf{r}_1 \ldots \mathbf{r}_k) \in \mathbb{R}^{d \times k}$ *be any orthonormal matrix such that* $\left|\bigcup_{j \in [t]} \mathrm{supp}(\mathbf{r}_j)\right| \leq s$ *and let* $\mathbf{u} \in \mathbb{R}^d$ *be a unit vector. Then for all* $r \in [d]$,

$$\left\|\mathbf{R}^\top \mathrm{top}_r(\mathbf{u})\right\|_2 \geq \left\|\mathbf{R}^\top \mathbf{u}\right\|_2$$

$$-\sqrt{\frac{s}{r}} \min\left\{1, \left(1 + \sqrt{\frac{s}{r}}\right)\left(1 - \|\mathbf{R}^\top \mathbf{u}\|_2^2\right)\right\}.$$

*Proof.* Let $\alpha := \|\mathbf{R}^\top \mathbf{u}\|_2 \in [0, 1]$ and $S := \bigcup_{j \in [k]} \mathrm{supp}(\mathbf{r}_j)$. If $\alpha = 0$, the claim is trivial since the right-hand side is $\leq 0$ while the left-hand side is $\geq 0$, so assume $\alpha > 0$.

Define

$$\mathbf{v}_\star := \frac{\mathbf{R}\mathbf{R}^\top \mathbf{u}}{\|\mathbf{R}^\top \mathbf{u}\|_2} = \frac{\mathbf{R}\mathbf{R}^\top \mathbf{u}}{\alpha}.$$

Since $\mathbf{R}^\top \mathbf{R} = \mathbf{I}_k$, $\mathbf{P} := \mathbf{R}\mathbf{R}^\top$ is the orthogonal projector onto $\mathrm{range}(\mathbf{R})$. Hence $\mathbf{v}_\star \in \mathrm{range}(\mathbf{R})$ and

$$\|\mathbf{v}_\star\|_2^2 = \frac{\|\mathbf{P}\mathbf{u}\|_2^2}{\alpha^2} = \frac{\mathbf{u}^\top \mathbf{P}^\top \mathbf{P}\mathbf{u}}{\alpha^2} = \frac{\mathbf{u}^\top \mathbf{P}\mathbf{u}}{\alpha^2} = \frac{\|\mathbf{V}^\top \mathbf{u}\|_2^2}{\alpha^2} = 1,$$

so $\mathbf{v}_\star$ is unit. Moreover, $\mathbf{P}\mathbf{u} = \mathbf{R}(\mathbf{R}^\top \mathbf{u}) = \sum_{j=1}^r (\mathbf{R}^\top \mathbf{u})_j \mathbf{r}_j$ is a linear combination of the columns of $\mathbf{R}$, so $\mathrm{supp}(\mathbf{P}\mathbf{u}) \subseteq \bigcup_{j=1}^r \mathrm{supp}(\mathbf{r}_j) = S$, and thus $\mathrm{supp}(\mathbf{v}_\star) \subseteq S$ and $|\mathrm{supp}(\mathbf{v}_\star)| \leq |S| \leq s$. Next,

$$\langle \mathbf{u}, \mathbf{v}_\star \rangle = \frac{\mathbf{u}^\top \mathbf{P}\mathbf{u}}{\alpha} = \frac{\|\mathbf{R}^\top \mathbf{u}\|_2^2}{\alpha} = \alpha, \text{ hence } 1 - \langle \mathbf{u}, \mathbf{v}_\star \rangle^2 = 1 - \alpha^2.$$

By Lemma 2 applied to $(\mathbf{u}, \mathbf{v}_\star)$ (with $|\mathrm{supp}(\mathbf{v}_\star)| \leq s$),

$$|\langle \mathrm{top}_r(\mathbf{u}), \mathbf{v}_\star \rangle| \geq |\langle \mathbf{u}, \mathbf{v}_\star \rangle| - \sqrt{\frac{s}{r}} \min\left\{ 1, \left(1 + \sqrt{\frac{s}{r}}\right)\left(1 - \langle \mathbf{u}, \mathbf{v}_\star \rangle^2\right)\right\}.$$

Substituting $|\langle \mathbf{u}, \mathbf{v}_\star \rangle| = \alpha$ and $1 - \langle \mathbf{u}, \mathbf{v}_\star \rangle^2 = 1 - \alpha^2$ yields

$$|\langle \mathrm{top}_r(\mathbf{u}), \mathbf{v}_\star \rangle| \geq \alpha - \sqrt{\frac{s}{r}} \min\left\{ 1, \left(1 + \sqrt{\frac{s}{r}}\right)\left(1 - \alpha^2\right)\right\}.$$

Finally, since $\mathbf{v}_\star \in \mathrm{range}(\mathbf{R})$ and $\|\mathbf{v}_\star\|_2 = 1$, we have

$$\|\mathbf{R}^\top \mathrm{top}_r(\mathbf{u})\|_2 = \max_{\substack{\mathbf{x} \in \mathrm{range}(\mathbf{R}) \\ \|\mathbf{x}\|_2 = 1}} |\langle \mathrm{top}_r(\mathbf{u}), \mathbf{x} \rangle| \geq |\langle \mathrm{top}_r(\mathbf{u}), \mathbf{v}_\star \rangle|.$$

Combining the last two displays and recalling $\alpha = \|\mathbf{R}^\top \mathbf{u}\|_2$ proves the claim. $\qquad\square$

**Lemma 5.** *Let $S \subseteq F \subseteq [d]$, and let $\mathbf{I}_F \in \{0,1\}^{d \times d}$ denote the diagonal matrix with $[\mathbf{I}_F]_{ii} := \mathbb{1}(i \in F)$. Let $\mathbf{x} \in \mathbb{R}^d, \|\mathbf{x}\|_2 = 1$ have support $F$ and let $\rho := \left\|\mathbf{V}_p^\top \mathbf{x}\right\|_2$. Define*

$$\mathbf{y} := \frac{\widehat{\mathbf{\Sigma}}_F \mathbf{x}}{\left\|\widehat{\mathbf{\Sigma}}_F \mathbf{x}\right\|_2}, \quad \mathbf{\Sigma}_F := \mathbf{I}_F \mathbf{\Sigma} \mathbf{I}_F, \quad \widehat{\mathbf{\Sigma}}_F := \mathbf{I}_F \widehat{\mathbf{\Sigma}} \mathbf{I}_F.$$

*If*

$$\epsilon_F := \frac{1}{\lambda_p(\mathbf{\Sigma})} \left\|\mathbf{\Sigma}_F - \widehat{\mathbf{\Sigma}}_F\right\|_{\mathrm{op}} \leq \frac{\lambda_p(\mathbf{\Sigma}) - \lambda_{p+1}(\mathbf{\Sigma})}{8\lambda_p(\mathbf{\Sigma})},$$

*then for $\kappa := \frac{\lambda_{p+1}(\mathbf{\Sigma})}{\lambda_p(\mathbf{\Sigma})}$, we may lower bound $\left\|\mathbf{V}_p^\top \mathbf{y}\right\|_2$ by*

$$\rho\left(1 + \frac{3(1 - \kappa)(1 - (\rho + \frac{2\epsilon_F}{1 - \kappa})^2)}{16}\right) - \frac{5\epsilon_F}{1 - \kappa}.$$

*Proof.* Let $\widehat{\mathbf{\Sigma}}_F := \widehat{\mathbf{V}} \widehat{\mathbf{\Lambda}} \widehat{\mathbf{V}}^\top + \widehat{\mathbf{V}}_\perp \widehat{\mathbf{\Lambda}}_\perp \widehat{\mathbf{V}}_\perp^\top$ where $\widehat{\mathbf{V}}$ denotes the top-$p$ eigenspace of $\widehat{\mathbf{\Sigma}}_F$, and $\widehat{\mathbf{V}}_\perp$ denotes the residual orthogonal eigenspace of $\widehat{\mathbf{\Sigma}}_F$ with $\hat{\kappa} := \frac{\lambda_{p+1}(\widehat{\mathbf{\Sigma}}_F)}{\lambda_p(\widehat{\mathbf{\Sigma}}_F)}$. Let $\mathbf{x} := \widehat{\mathbf{V}}\boldsymbol{\alpha} + \widehat{\mathbf{V}}_\perp \boldsymbol{\beta}$ where $\|\boldsymbol{\alpha}\|_2^2 + \|\boldsymbol{\beta}\|_2^2 = 1$, and let $\eta := \|\boldsymbol{\alpha}\|_2$. Then,

$$\left\|\widehat{\mathbf{V}}\widehat{\mathbf{V}}^\top \mathbf{y}\right\|_2 = \left\|\widehat{\mathbf{V}}^\top \mathbf{y}\right\|_2 = \left|\frac{\widehat{\mathbf{V}}^\top \widehat{\mathbf{\Sigma}}_F \mathbf{x}}{\left\|\widehat{\mathbf{\Sigma}}_F \mathbf{x}\right\|_2}\right| = \frac{\left\|\widehat{\mathbf{\Lambda}}\boldsymbol{\alpha}\right\|_2}{\sqrt{\left\|\widehat{\mathbf{\Lambda}}\boldsymbol{\alpha}\right\|_2^2 + \left\|\widehat{\mathbf{\Lambda}}_\perp\boldsymbol{\beta}\right\|_2^2}}$$

$$\geq \frac{\lambda_p\left(\widehat{\mathbf{\Sigma}}_F\right)\eta}{\sqrt{\lambda_p\left(\widehat{\mathbf{\Sigma}}_F\right)^2 \eta^2 + \lambda_{p+1}\left(\widehat{\mathbf{\Sigma}}_F\right)^2 (1 - \eta^2)}} \qquad (16)$$

$$= \frac{\eta}{\sqrt{1 - (1 - \hat{\kappa}^2)(1 - \eta^2)}} \geq \eta\left(1 + \frac{(1 - \hat{\kappa}^2)(1 - \eta^2)}{2}\right)$$

where the last inequality follows due to $\frac{1}{\sqrt{1-z}} \geq 1 + \frac{z}{2}$ for $z \in (0, 1]$. Let $\mathbf{\Lambda}_p, \mathbf{\Lambda}_{p,\perp}$ denote the top-$p$ and the corresponding orthogonal eigenspaces of $\mathbf{\Sigma}$. Then since $S \subseteq F$,

$$\mathbf{\Sigma}_F := \mathbf{I}_F \left(\mathbf{V}_p \mathbf{\Lambda} \mathbf{V}_p^\top + \mathbf{V}_{p,\perp} \mathbf{\Lambda}_{p,\perp} \mathbf{V}_{p,\perp}^\top\right) \mathbf{I}_F = \mathbf{V}_p \mathbf{\Lambda} \mathbf{V}_p^\top + \mathbf{I}_F \mathbf{V}_{p,\perp} \mathbf{\Lambda}_{p,\perp} \mathbf{V}_{p,\perp}^\top \mathbf{I}_F$$

which implies that $\mathbf{V}_p$ represents the top-$p$ eigenspace of $\mathbf{\Sigma}_F$ with $\lambda_p(\mathbf{\Sigma}_F) = \lambda_p(\mathbf{\Sigma})$ and $\lambda_{p+1}(\mathbf{\Sigma}_F) \leq \lambda_{p+1}(\mathbf{\Sigma}) < \lambda_p(\mathbf{\Sigma})$. For convenience of notation, denote $\forall i \in [d]$, $\lambda_i(\mathbf{\Sigma}) \equiv \lambda_i$. Then,

$$
\begin{aligned}
1 - \hat{\kappa}^2 = (1 + \hat{\kappa})(1 - \hat{\kappa}) \geq 1 - \hat{\kappa} &= \frac{\lambda_p(\widehat{\mathbf{\Sigma}}_F) - \lambda_{p+1}(\widehat{\mathbf{\Sigma}}_F)}{\lambda_p(\widehat{\mathbf{\Sigma}}_F)} \\
&\geq \frac{\lambda_p(\mathbf{\Sigma}_F) - \epsilon_F \lambda_p(\mathbf{\Sigma}) - \lambda_{p+1}(\mathbf{\Sigma}_F) - \epsilon_F \lambda_p(\mathbf{\Sigma})}{\lambda_p(\mathbf{\Sigma}_F) + \epsilon_F \lambda_p(\mathbf{\Sigma})}, \text{ using Weyl's inequality} \\
&= \frac{\lambda_p(\mathbf{\Sigma}_F) - \lambda_{p+1}(\mathbf{\Sigma}_F) - 2\epsilon_F \lambda_p(\mathbf{\Sigma})}{\lambda_p(\mathbf{\Sigma}_F) + \epsilon_F \lambda_p(\mathbf{\Sigma})} \\
&= \frac{1 - \lambda_{p+1}(\mathbf{\Sigma}_F)/\lambda_p(\mathbf{\Sigma}_F) - 2\epsilon_F}{1 + \epsilon_F}, \text{ since } \lambda_p(\mathbf{\Sigma}_F) = \lambda_p(\mathbf{\Sigma}) \\
&\geq \frac{1 - \lambda_{p+1}/\lambda_p - 2\epsilon_F}{1 + \epsilon_F}, \text{ since } \frac{\lambda_{p+1}(\mathbf{\Sigma}_F)}{\lambda_p(\mathbf{\Sigma}_F)} \leq \frac{\lambda_{p+1}}{\lambda_p}, \\
&\geq \frac{1 - \lambda_{p+1}/\lambda_p - 0.25(\lambda_p - \lambda_{p+1})/\lambda_p}{1 + \epsilon_F}, \text{ using } \epsilon_F < \frac{\lambda_p - \lambda_{p+1}}{8\lambda_p} \\
&= \frac{3}{4(1 + \epsilon_F)} \frac{\lambda_p - \lambda_{p+1}}{\lambda_p} \geq \frac{3(1 - \kappa)}{8}, \text{ since } \epsilon_F < 1.
\end{aligned}
\tag{17}
$$

Using an application of Wedin's theorem (in an alternate form stated in Lemma 16) along with the bound on $\epsilon_F$, we have $\|\widehat{\mathbf{V}}\widehat{\mathbf{V}}^\top - \mathbf{V}_p \mathbf{V}_p^\top\|_{\text{op}} \leq \frac{2\epsilon_F \lambda_p(\mathbf{\Sigma})}{\lambda_p(\mathbf{\Sigma})(1 - \kappa)} = \frac{2\epsilon_F}{1 - \kappa}$. Then,

$$
\rho - \frac{2\epsilon_F}{1 - \kappa} \leq \eta \leq \rho + \frac{2\epsilon_F}{1 - \kappa}.
$$

Therefore, combining (16), (17), and the above bound,

$$
\begin{aligned}
\left\|\mathbf{V}_p^\top \mathbf{y}\right\|_2 = \left\|\mathbf{V}_p \mathbf{V}_p^\top \mathbf{y}\right\|_2 &\geq \left\|\widehat{\mathbf{V}}\widehat{\mathbf{V}}^\top \mathbf{y}\right\|_2 - \frac{2\epsilon_F}{1 - \kappa} \\
&\geq \left(\rho - \frac{2\epsilon_F}{1 - \kappa}\right)\left(1 + \frac{(1 - \hat{\kappa}^2)(1 - (\rho + \frac{2\epsilon_F}{1 - \kappa})^2)}{2}\right) - \frac{2\epsilon_F}{(1 - \kappa)} \\
&\geq \rho\left(1 + \frac{(1 - \hat{\kappa}^2)(1 - (\rho + \frac{2\epsilon_F}{1 - \kappa})^2)}{2}\right) - \frac{5\epsilon_F}{1 - \kappa} \\
&= \rho\left(1 + \frac{3(1 - \kappa)(1 - (\rho + \frac{2\epsilon_F}{1 - \kappa})^2)}{16}\right) - \frac{5\epsilon_F}{1 - \kappa}.
\end{aligned}
$$

$\square$

**Lemma 6.** *Suppose that under Model 3, we have in Algorithm 1 that for some $i \in [d]$ and $t \in [T]$,*

$$
\psi_{t-1}^{(i)} \in \left[\sqrt{\frac{p}{s}}, \frac{1}{\sqrt{2}}\right].
$$

*Then, for $\delta \in (0, 1)$, if we take*

$$
r = \Omega\left(\frac{s^2 k^2}{\beta^2}\right), \quad n = \Omega\left(\left(\frac{\sigma^2}{\lambda_p(\mathbf{\Sigma})}\right)^2 \cdot \frac{s^3 k^6}{\beta^6} \log\left(\frac{d}{\delta}\right)\right)
$$

*for appropriate constants, with probability at least $1 - \delta$, we have $\psi_t^{(i)} \geq (1 + \frac{\beta}{16k})\psi_{t-1}^{(i)}$.*

*Proof.* Recall that $S := \bigcup_{j \in [k]} \text{supp}(\mathbf{v}_j)$. Define the sets, for all $t \in [T]$ and $i \in [d]$,

$$
F_t^{(i)} := S \cup \text{supp}(\mathbf{u}_t^{(i)}) \cup \text{supp}(\mathbf{u}_{t-1}^{(i)}).
\tag{18}
$$

Then, $|F_t^{(i)}| \leq |\text{supp}(\mathbf{u}_t^{(i)})| + |\text{supp}(\mathbf{u}_{t-1}^{(i)})| + |S| \leq 3r$. For any $F \subseteq [d]$, define $\mathbf{I}_F$ as in Lemma 5. Next, for all $t \in [T]$ and $i \in [d]$, define

$$\widehat{\mathbf{\Sigma}}_{F_t^{(i)}} := \mathbf{I}_{F_t^{(i)}} \widehat{\mathbf{\Sigma}} \mathbf{I}_{F_t^{(i)}}, \quad \mathbf{\Sigma}_{F_t^{(i)}} := \mathbf{I}_{F_t^{(i)}} \mathbf{\Sigma} \mathbf{I}_{F_t^{(i)}}, \quad \epsilon_{F_t^{(i)}} := \frac{\left\| \widehat{\mathbf{\Sigma}}_{F_t^i} - \mathbf{\Sigma}_{F_t^{(i)}} \right\|_{\text{op}}}{\lambda_p(\mathbf{\Sigma}_{F_t^{(i)}})}, \quad \kappa := \frac{\lambda_{p+1}(\mathbf{\Sigma})}{\lambda_p(\mathbf{\Sigma})}. \tag{19}$$

From the above definitions and from Algorithm 1,

$$\mathbf{u}_t^{(i)} = \frac{\text{top}_r\left( \widehat{\mathbf{\Sigma}} \mathbf{u}_{t-1}^{(i)} \right)}{\left\| \text{top}_r\left( \widehat{\mathbf{\Sigma}} \mathbf{u}_{t-1}^{(i)} \right) \right\|_2} = \frac{\text{top}_r\left( \widehat{\mathbf{\Sigma}}_{F_t^{(i)}} \mathbf{u}_{t-1}^{(i)} \right)}{\left\| \text{top}_r\left( \widehat{\mathbf{\Sigma}}_{F_t^{(i)}} \mathbf{u}_{t-1}^{(i)} \right) \right\|_2} = \frac{\text{top}_r\left( \mathbf{y}_{t-1}^{(i)} \right)}{\left\| \text{top}_r\left( \mathbf{y}_{t-1}^{(i)} \right) \right\|_2}$$

where $\mathbf{y}_{t-1}^{(i)} := \widehat{\mathbf{\Sigma}}_{F_t^{(i)}} \mathbf{u}_{t-1}^{(i)} \cdot \| \widehat{\mathbf{\Sigma}}_{F_t^{(i)}} \mathbf{u}_{t-1}^{(i)} \|_2^{-1}$. Then we have

$$\psi_t^{(i)} = \left\| \mathbf{V}_p^\top \mathbf{u}_t^{(i)} \right\|_2 = \left\| \mathbf{V}_p^\top \frac{\text{top}_r\left( \mathbf{y}_{t-1}^{(i)} \right)}{\left\| \text{top}_r\left( \mathbf{y}_{t-1}^{(i)} \right) \right\|} \right\|_2 \geq \left\| \mathbf{V}_p^\top \text{top}_r\left( \mathbf{y}_{t-1}^{(i)} \right) \right\|_2$$

$$\overset{(a)}{\geq} \left\| \mathbf{V}_p^\top \mathbf{y}_{t-1}^{(i)} \right\|_2 - \sqrt{\frac{s}{r}} \min\left\{ 1, \left( 1 + \sqrt{\frac{s}{r}} \right) \left( 1 - \left\| \mathbf{V}_p^\top \mathbf{y}_{t-1}^{(i)} \right\|_2^2 \right) \right\}$$

$$\geq \left\| \mathbf{V}_p^\top \mathbf{y}_{t-1}^{(i)} \right\|_2 - \sqrt{\frac{s}{r}}$$

$$\overset{(b)}{\geq} \psi_{t-1}^{(i)} \left( 1 + \frac{3(1-\kappa)\left( 1 - \left( \psi_{t-1}^{(i)} + 2\epsilon_{F_t^{(i)}}/(1-\kappa) \right)^2 \right)}{16} \right) - \frac{5\epsilon_{F_t^{(i)}}}{1-\kappa} - \sqrt{\frac{s}{r}}, \tag{20}$$

where we used Lemma 3 in $(a)$, and Lemma 5 in $(b)$. Note that using Corollary 1, for the stated range of $n$, we have for all $i \in [d]$ and $t \in [T]$ that $\epsilon_{F_t^{(i)}} \leq \frac{1}{8} \cdot \frac{\lambda_p(\mathbf{\Sigma}) - \lambda_{p+1}(\mathbf{\Sigma})}{\lambda_p(\mathbf{\Sigma})}$, and therefore Lemma 5 is applicable. Further, we note that for the stated range of $r$ and $n$, we have

$$\max\left\{ \sqrt{\frac{s}{r}}, \frac{6\epsilon_{F_t^{(i)}}}{1-\kappa} \right\} \leq \frac{1-\kappa}{64} \sqrt{\frac{p}{s}} \leq \frac{1-\kappa}{64} \psi_t^{(i)}.$$

Therefore, continuing from (20), we have

$$\psi_t^{(i)} \geq \psi_{t-1}^{(i)} \left( 1 + \frac{3(1-\kappa)(1 - (\psi_{t-1}^{(i)} + 2\epsilon_{F_t^{(i)}}/(1-\kappa))^2)}{16} \right) - \frac{5\epsilon_{F_t^{(i)}}}{1-\kappa} - \sqrt{\frac{s}{r}}$$

$$\geq \psi_{t-1}^{(i)} \left( 1 + \frac{3(1-\kappa)(1 - (\psi_{t-1}^{(i)})^2)}{16} \right) - \frac{6\epsilon_{F_t^{(i)}}}{1-\kappa} - \sqrt{\frac{s}{r}} \tag{21}$$

$$\geq \psi_{t-1}^{(i)} \left( 1 + \frac{3(1-\kappa)}{32} \right) - \frac{6\epsilon_{F_t^{(i)}}}{1-\kappa} - \sqrt{\frac{s}{r}}, \quad \text{using } \psi_{t-1}^{(i)} \leq \frac{1}{\sqrt{2}}$$

$$\geq \psi_{t-1}^{(i)} \left( 1 + \frac{3(1-\kappa)}{32} \right) - \frac{2(1-\kappa)}{64} \psi_t^{(i)} = \psi_{t-1}^{(i)} \left( 1 + \frac{(1-\kappa)}{16} \right). \tag{22}$$

The claim then follows by noting that under (4), $1 - \kappa = 1 - \frac{\lambda_{p+1}(\mathbf{\Sigma})}{\lambda_p(\mathbf{\Sigma})} \geq 1 - (1 - \frac{\beta}{k}) = \frac{\beta}{k}$. $\qquad \square$

**Lemma 7.** *Suppose that under Model 3, we have in Algorithm 1 that for some $i \in [d]$ and $t \in [T]$,*

$$\psi_{t-1}^{(i)} \in \left[ \frac{1}{\sqrt{2}}, \sqrt{1-\Delta} \right].$$

*Then, for $\delta \in (0,1)$, $\Delta \in (0, \frac{1}{2})$, if we take*

$$r = \Omega\left(\frac{sk^2}{\Delta^2\beta^2}\right), \quad n = \Omega\left(\left(\frac{\sigma^2}{\lambda_p(\mathbf{\Sigma})}\right)^2 \cdot \frac{s^2k^6}{\Delta^4\beta^6}\log\left(\frac{d}{\delta}\right)\right)$$

*for appropriate constants, with probability at least $1 - \delta$, we have $\widetilde{\psi}_t^{(i)} \leq (1 - \frac{\beta}{8k})\widetilde{\psi}_{t-1}^{(i)}$ for*

$$\widetilde{\psi}_t^{(i)} := 1 - (\psi_t^{(i)})^2.$$

*Proof.* We follow the steps and notation from Lemma 6, differing primarily after (21). As before,

$$\psi_t^{(i)} \geq \psi_{t-1}^{(i)}\left(1 + \frac{3(1-\kappa)(1-(\psi_{t-1}^{(i)})^2)}{16}\right) - \frac{6\epsilon_{F_t^{(i)}}}{1-\kappa} - \sqrt{\frac{s}{r}}. \tag{23}$$

By definition, $\widetilde{\psi}_t^{(i)} := 1 - (\psi_t^{(i)})^2$. Then for appropriately large

$$n = \Omega\left(\left(\frac{\sigma^2}{\lambda_p(\mathbf{\Sigma})} \cdot \frac{1}{(1-\kappa)^2} \cdot \frac{1}{\Delta}\right)^2 rs\log\left(\frac{d}{\delta}\right)\right), \quad r = \Omega\left(\frac{s}{\Delta^2(1-\kappa)^2}\right),$$

we have by using Corollary 1 and $\widetilde{\psi}_t^{(i)} \geq \Delta$,

$$\max\left\{\sqrt{\frac{s}{r}}, \frac{6\epsilon_{F_t^{(i)}}}{1-\kappa}\right\} \leq \frac{(1-\kappa)\Delta}{96} \leq \frac{(1-\kappa)}{96}\widetilde{\psi}_t^{(i)}.$$

Squaring both sides of (23) and subtracting from 1, along with using $1 > \sqrt{1-\Delta} \geq \psi_t^{(i)} \geq 1/\sqrt{2}$,

$$\widetilde{\psi}_t^{(i)} \leq \widetilde{\psi}_{t-1}^{(i)} - 2(\psi_{t-1}^{(i)})^2 \frac{3(1-\kappa)\widetilde{\psi}_t^{(i)}}{16} + 2\psi_{t-1}^{(i)}\frac{19}{16}\left(\sqrt{\frac{s}{r}} + \frac{5\epsilon_{F_t^{(i)}}}{1-\kappa}\right)$$

$$\leq \widetilde{\psi}_t^{(i)}\left(1 - \frac{3(1-\kappa)}{16}\right) + 3\left(\sqrt{\frac{s}{r}} + \frac{5\epsilon_{F_t^{(i)}}}{1-\kappa}\right)$$

$$\leq \widetilde{\psi}_{t-1}^{(i)}\left(1 - \frac{3(1-\kappa)}{16}\right) + 3\frac{2(1-\kappa)}{96}\widetilde{\psi}_{t-1}^{(i)}$$

$$\leq \widetilde{\psi}_{t-1}^{(i)}\left(1 - \frac{2(1-\kappa)}{16}\right) = \widetilde{\psi}_{t-1}^{(i)}\left(1 - \frac{(1-\kappa)}{8}\right). \tag{24}$$

The claim then follows by noting that under (4), $1 - \kappa = 1 - \frac{\lambda_{p+1}(\mathbf{\Sigma})}{\lambda_p(\mathbf{\Sigma})} \geq 1 - (1 - \frac{\beta}{k}) = \frac{\beta}{k}$. $\qquad\square$

**Proposition 1.** *Let $\delta \in (0,1)$, $\epsilon \in (0, \frac{1}{2})$, and under Model 3, assume that*

$$n = \Omega\left(\left(\frac{\sigma^2}{\lambda_k(\mathbf{\Sigma})}\right)^2\left(\frac{s^3k^6}{\beta^6} + \frac{s^2k^6}{\epsilon^4\beta^6}\right)\log\left(\frac{d}{\delta}\right)\right),$$

$$r = \Omega\left(\frac{s^2k^2}{\beta^2} + \frac{sk^2}{\beta^2\epsilon^2}\right), \quad T = \Omega\left(\frac{k\log\left(\frac{s}{\epsilon}\right)}{\epsilon}\right)$$

*for appropriate constants. Then, in time $O\left(nd^2T\right)$, Algorithm 1 returns an $r$-sparse unit vector $\mathbf{u}$ such that with probability at least $1 - \delta$, $\langle\mathbf{u}, \mathbf{\Sigma}\mathbf{u}\rangle \geq (1 - \epsilon - \beta)\lambda_1(\mathbf{\Sigma})$.*

*Proof.* The time complexity immediately follows from Lemma 4. We now provide the convergence argument. Let $S := \bigcup_{i\in[k]}\text{supp}(\mathbf{v}_i)$. Since $\mathbf{V}_p$ has orthonormal columns, $\sum_{i\in S}\|\mathbf{V}_p^\top\mathbf{e}_i\|_2^2 = \|\mathbf{V}_p\|_{\text{F}}^2 = p$, and hence there exists $i^\star \in S$ such that

$$\psi_0^{(i^\star)} := \|\mathbf{V}_p^\top\mathbf{e}_{i^\star}\|_2 \geq \sqrt{\frac{p}{s}}.$$

For the stated ranges of $r$ and $n$, Lemma 6 and Lemma 7 both apply with probability $\geq 1 - \frac{\delta}{2}$, and we avoid a union bound as every iteration is conditioning on the same event. Hence, as long as $\psi_t^{(i^\star)} \in [\sqrt{p/s}, 1/\sqrt{2}]$, using Lemma 6, after $T_1 = \Omega(\frac{k \log(s)}{\beta})$ iterations, we obtain $\psi_{T_1}^{(i_\star)} \geq \frac{1}{\sqrt{2}}$. Next, using Lemma 7 with $\Delta \leftarrow \frac{\epsilon}{3}$, we have, for $\psi_t^{(i_\star)} \in [1/\sqrt{2}, \sqrt{1-\Delta}]$, after $T_2 = \Omega(\frac{k \log \frac{1}{\Delta}}{\beta})$ iterations, we obtain $\widetilde{\psi}_{T_1+T_2}^{(i_\star)} \leq \Delta$, or equivalently, $\psi_{T_1+T_2}^{(i_\star)} \geq \sqrt{1-\Delta}$. Thus, for the output $\mathbf{u}^{(i^\star)} := \mathbf{u}_T^{(i^\star)}$,

$$\left\| \mathbf{V}_p^\top \mathbf{u}^{(i_\star)} \right\|_2^2 \geq 1 - \frac{\epsilon}{3} \tag{25}$$

with probability at least $1 - \frac{\delta}{2}$ for the stated range of $n$. Denote this event by $\mathcal{E}$.

Then, since $\mathbf{u}^{(i_\star)}$ is $r$-sparse,

$$
\begin{aligned}
(\mathbf{u}^{(i_\star)})^\top \widehat{\boldsymbol{\Sigma}} \mathbf{u}^{(i_\star)} &\geq (\mathbf{u}^{(i_\star)})^\top \boldsymbol{\Sigma} \mathbf{u}^{(i_\star)} - \max_{F \subseteq [d], |F| \leq r} \left\| \widehat{\boldsymbol{\Sigma}}_{F \times F} - \boldsymbol{\Sigma}_{F \times F} \right\|_{\mathrm{op}} \\
&= (\mathbf{u}^{(i_\star)})^\top (\mathbf{V}_p \boldsymbol{\Lambda} \mathbf{V}_p^\top + \mathbf{V}_{p,\perp} \boldsymbol{\Lambda}_{p,\perp} \mathbf{V}_{p,\perp}^\top) \mathbf{u}^{(i_\star)} - \max_{F \subseteq [d], |F| \leq r} \left\| \widehat{\boldsymbol{\Sigma}}_{F \times F} - \boldsymbol{\Sigma}_{F \times F} \right\|_{\mathrm{op}} \\
&\geq (\mathbf{V}_p^\top \mathbf{u}^{(i_\star)})^\top \boldsymbol{\Lambda}_p (\mathbf{V}_p^\top \mathbf{u}^{(i_\star)}) - \max_{F \subseteq [d], |F| \leq r} \left\| \widehat{\boldsymbol{\Sigma}}_{F \times F} - \boldsymbol{\Sigma}_{F \times F} \right\|_{\mathrm{op}} \\
&\geq \lambda_p(\boldsymbol{\Sigma}) \left\| \mathbf{V}_p^\top \mathbf{u}^{(i_\star)} \right\|_2^2 - \max_{F \subseteq [d], |F| \leq r} \left\| \widehat{\boldsymbol{\Sigma}}_{F \times F} - \boldsymbol{\Sigma}_{F \times F} \right\|_{\mathrm{op}}.
\end{aligned}
\tag{26}
$$

Using Corollary 1, for sufficiently large

$$n = \Omega\left( \left( \frac{\sigma^2}{\lambda_p(\boldsymbol{\Sigma})} \right)^2 \frac{r}{\epsilon^2} \log\left( \frac{d}{\delta} \right) \right),$$

with probability at least $1 - \frac{\delta}{2}$, $\max_{F \subseteq [d], |F| \leq r} \|\widehat{\boldsymbol{\Sigma}}_{F \times F} - \boldsymbol{\Sigma}_{F \times F}\|_{\mathrm{op}} \leq \frac{\epsilon}{3} \lambda_p(\boldsymbol{\Sigma})$. Condition on this and $\mathcal{E}$ in the rest of the proof, giving the failure probability. Under these events, (25) and (26) yield

$$(\mathbf{u}^{(i_\star)})^\top \widehat{\boldsymbol{\Sigma}} \mathbf{u}^{(i_\star)} \geq \left( 1 - \frac{2\epsilon}{3} \right) \lambda_p(\boldsymbol{\Sigma}).$$

Finally, let $j := \mathrm{argmax}_{i \in [d]} (\mathbf{u}^{(i)})^\top \widehat{\boldsymbol{\Sigma}} \mathbf{u}^{(i)}$ be the selected index on Line 9. Then,

$$(\mathbf{u}^{(j)})^\top \widehat{\boldsymbol{\Sigma}} \mathbf{u}^{(j)} \geq (\mathbf{u}^{(i_\star)})^\top \widehat{\boldsymbol{\Sigma}} \mathbf{u}^{(i_\star)} \geq \left( 1 - \frac{2\epsilon}{3} \right) \lambda_p(\boldsymbol{\Sigma}) \geq \left( 1 - \frac{2\epsilon}{3} - \beta \right) \lambda_1(\boldsymbol{\Sigma}) \tag{27}$$

where in the last inequality we used the definition of $p$ in (4). The result follows from $r$-sparsity of $\mathbf{u}^{(j)}$, so that $(\mathbf{u}^{(j)})^\top \boldsymbol{\Sigma} \mathbf{u}^{(j)} \geq (\mathbf{u}^{(j)})^\top \widehat{\boldsymbol{\Sigma}} \mathbf{u}^{(j)} - \frac{\epsilon}{3} \lambda_p(\boldsymbol{\Sigma})$ under our earlier conditioning event. $\square$

**Lemma 9.** *Let* $(\Delta, \gamma) \in (0,1)^2$ *and* $d \geq 4$. *There exists* $\boldsymbol{\Sigma} \in \mathbb{S}_{\succeq \mathbf{0}}^{d \times d}$ *such that* $(1 - \gamma)\lambda_2(\boldsymbol{\Sigma}) \geq \lambda_3(\boldsymbol{\Sigma})$, *and* $\mathbf{v}_1, \mathbf{v}_2$, *the top two eigenvectors of* $\boldsymbol{\Sigma}$, *are both supported on a 2-sparse set, yet for a 3-sparse unit* $\mathbf{u} \in \mathbb{R}^d$ *with* $\langle \mathbf{u}, \mathbf{v}_1 \rangle^2 \geq 1 - \Delta$, *if* $\mathbf{P} := \mathbf{I}_d - \mathbf{u}\mathbf{u}^\top$, *then* $|\mathrm{supp}(\mathbf{v}_1(\mathbf{P}\boldsymbol{\Sigma}\mathbf{P}))| = d$.

*Proof.* Throughout this proof, let $\mathbf{v}_1 = \frac{1}{\sqrt{2}}(\mathbf{e}_1 + \mathbf{e}_2)$, $\mathbf{v}_2 = \frac{1}{\sqrt{2}}(\mathbf{e}_1 - \mathbf{e}_2)$, and define

$$\mathbf{q} := \frac{1}{\sqrt{d-3}} \sum_{i=4}^d \mathbf{e}_i, \quad \mathbf{w} := \frac{1}{\sqrt{2}}(\mathbf{e}_3 + \mathbf{q}).$$

We also define

$$\boldsymbol{\Sigma} := \frac{1}{\Delta} \mathbf{v}_1 \mathbf{v}_1^\top + \mathbf{v}_2 \mathbf{v}_2^\top + (1 - \gamma)\mathbf{w}\mathbf{w}^\top, \quad \mathbf{P} := \mathbf{I}_d - \mathbf{u}\mathbf{u}^\top \text{ where } \mathbf{u} := \sqrt{1 - \Delta}\mathbf{v}_1 + \sqrt{\Delta}\mathbf{e}_3.$$

It is clear that all of the conditions of the lemma statement are now met, and what remains to be shown is that the support of $\mathbf{v}_1(\mathbf{P}\boldsymbol{\Sigma}\mathbf{P})$ has large cardinality. For convenience we also let

$$\mathbf{t} := \sqrt{\Delta}\mathbf{v}_1 - \sqrt{1 - \Delta}\mathbf{e}_3.$$

Now observe that

$$\mathbf{P}\boldsymbol{\Sigma}\mathbf{P} = \frac{1}{\Delta}\mathbf{P}\mathbf{v}_1\mathbf{v}_1^\top\mathbf{P} + \mathbf{v}_2\mathbf{v}_2^\top + (1 - \gamma)\mathbf{P}\mathbf{w}\mathbf{w}^\top\mathbf{P},$$

so that $\mathrm{range}(\mathbf{P}\boldsymbol{\Sigma}\mathbf{P}) = \mathrm{span}\{\mathbf{P}\mathbf{v}_1, \mathbf{v}_2, \mathbf{P}\mathbf{w}\}$. It is straightforward to check that

$$\mathbf{P}\mathbf{v}_1 = \sqrt{\Delta}\mathbf{t}, \quad \mathbf{P}\mathbf{w} = \mathbf{w} - \sqrt{\frac{\Delta}{2}}\mathbf{u} = \frac{1}{\sqrt{2}}\left(\mathbf{q} - \sqrt{1 - \Delta}\mathbf{t}\right),$$

and that $\mathbf{t} \perp \mathbf{q}$ are unit vectors, so equivalently $\mathrm{range}(\mathbf{P}\boldsymbol{\Sigma}\mathbf{P}) = \mathrm{span}\{\mathbf{t}, \mathbf{v}_2, \mathbf{q}\}$. This same calculation also shows that $\mathrm{span}\{\mathbf{t}, \mathbf{q}\}$ and $\mathrm{span}\{\mathbf{v}_2\}$ are invariant subspaces with respect to $\mathbf{P}\boldsymbol{\Sigma}\mathbf{P}$. Thus, the top eigenvector is either $\mathbf{v}_2$, or a vector in $\mathrm{span}\{\mathbf{t}, \mathbf{q}\}$.

We next compute the compression of $\mathbf{P}\boldsymbol{\Sigma}\mathbf{P}$ in the basis $\mathrm{span}\{\mathbf{t}, \mathbf{q}\}$: because $\mathbf{t} \perp \mathbf{u}$ and $\mathbf{q} \perp \mathbf{u}$,

$$\mathbf{t}^\top\mathbf{P}\boldsymbol{\Sigma}\mathbf{P}\mathbf{t} = \mathbf{t}^\top\boldsymbol{\Sigma}\mathbf{t} = \frac{1}{\Delta}\langle\mathbf{v}_1, \mathbf{t}\rangle^2 + (1 - \gamma)\langle\mathbf{w}, \mathbf{t}\rangle^2 = 1 + \frac{(1 - \gamma)(1 - \Delta)}{2},$$

$$\mathbf{t}^\top\mathbf{P}\boldsymbol{\Sigma}\mathbf{P}\mathbf{q} = \mathbf{t}^\top\boldsymbol{\Sigma}\mathbf{q} = (1 - \gamma)\mathbf{t}^\top\mathbf{w}\mathbf{w}^\top\mathbf{q} = -\frac{(1 - \gamma)\sqrt{1 - \Delta}}{2},$$

$$\mathbf{q}^\top\mathbf{P}\boldsymbol{\Sigma}\mathbf{P}\mathbf{q} = \mathbf{q}^\top\boldsymbol{\Sigma}\mathbf{q} = (1 - \gamma)\langle\mathbf{w}, \mathbf{q}\rangle^2 = \frac{1 - \gamma}{2}.$$

Thus, the compression of $\mathbf{P}\boldsymbol{\Sigma}\mathbf{P}$ in this invariant subspace is

$$\mathbf{C} := \begin{pmatrix} 1 + \frac{(1-\gamma)(1-\Delta)}{2} & -\frac{(1-\gamma)\sqrt{1-\Delta}}{2} \\ -\frac{(1-\gamma)\sqrt{1-\Delta}}{2} & \frac{1-\gamma}{2} \end{pmatrix}.$$

The top eigenvector of $\mathbf{P}\boldsymbol{\Sigma}\mathbf{P}$ in this subspace has nonzero mass on both the $\mathbf{t}$ and $\mathbf{q}$ directions (because $\mathbf{C}$ is not diagonal), so it has density $d$, and its corresponding eigenvalue is clearly $> 1$ as witnessed by the top-left entry, so it is also the top eigenvector overall (as $\mathbf{P}\boldsymbol{\Sigma}\mathbf{P}\mathbf{v}_2 = \mathbf{v}_2$). $\qquad\square$

## D. Deferred proofs

In Appendix B.3, we use the following helper result.

**Lemma 14.** *For any $d \in \mathbb{N}$, there is an orthonormal basis $\{\mathbf{u}_i\}_{i\in[d]}$ such that $\mathbf{u}_1 = \frac{1}{\sqrt{d}}\mathbf{1}_d$, and for all $i \in [d] \setminus \{1\}$, we have that $\langle\mathbf{u}_i, \mathbf{e}_1\rangle = \frac{1}{\sqrt{d}}$.*

*Proof.* Let $\mathbf{u} := \mathbf{u}_1$ and let $H \subseteq \mathbb{R}^d$ be the subspace orthogonal to $\mathbf{u}$. Also, let $\mathbf{w} = \mathbf{e}_1 - \frac{1}{\sqrt{d}}\mathbf{u} \in H$ so that $\|\mathbf{w}\|_2^2 = \frac{d-1}{d}$. Then, the lemma statement demands that the $\{\mathbf{u}_i\}_{i\in[d]\setminus\{1\}}$ are an orthonormal basis of $H$ such that $\langle\mathbf{w}, \mathbf{u}_i\rangle = \frac{1}{\sqrt{d}}$ for all $i \in [d] \setminus \{1\}$. To construct such a basis, let $\mathbf{V} \in \mathbb{R}^{d\times(d-1)}$ be an arbitrary orthonormal matrix whose columns span $H$, and define

$$\mathbf{x} := \mathbf{V}^\top\left(\frac{\mathbf{w}}{\|\mathbf{w}\|_2}\right), \quad \mathbf{t} := \frac{1}{\sqrt{d-1}}\mathbf{1}_{d-1}.$$

Let $\mathbf{Q} \in \mathbb{R}^{(d-1)\times(d-1)}$ be the orthonormal matrix guaranteed by Lemma 15 to satisfy $\mathbf{Q}\mathbf{x} = \mathbf{t}$. We claim it suffices to take $\mathbf{u}_i = \mathbf{V}\mathbf{Q}_{i-1:}$ for all $i \in [d] \setminus \{1\}$ where $\mathbf{Q}_{i-1:}$ is the $i^{\text{th}}$ row of $\mathbf{Q}$.

To check this, it is clear that all $\mathbf{u}_i$ have unit norm and are pairwise orthogonal, by orthogonality of $\mathbf{V}$ and $\mathbf{Q}$. Moreover, since $\mathbf{V}^\top\mathbf{u}_1 = \mathbf{0}_{d-1}$ these vectors are also all orthogonal to $\mathbf{v}_1$. Finally,

$$\langle\mathbf{w}, \mathbf{u}_i\rangle = \mathbf{w}^\top\mathbf{V}\mathbf{Q}^\top\mathbf{e}_i = \|\mathbf{w}\|_2\mathbf{x}^\top\mathbf{Q}^\top\mathbf{e}_i = \|\mathbf{w}\|_2\mathbf{t}^\top\mathbf{e}_i = \frac{1}{\sqrt{d}}, \text{ for all } i \in [d] \setminus \{1\}.$$

$\qquad\square$

The proof of Lemma 14 used the following claim.

**Lemma 15.** *Let* $\mathbf{x}, \mathbf{t} \in \mathbb{R}^d$ *be arbitrary unit vectors. Then, there exists an orthonormal matrix* $\mathbf{Q} \in \mathbb{R}^{d \times d}$ *such that* $\mathbf{Q}\mathbf{x} = \mathbf{t}$.

*Proof.* If $\mathbf{x} = \mathbf{t}$ the claim is trivial. Otherwise, choose

$$\mathbf{Q} := \mathbf{I}_d - 2 \frac{\mathbf{u}\mathbf{u}^\top}{\|\mathbf{u}\|_2^2}, \text{ where } \mathbf{u} := \mathbf{x} - \mathbf{t}.$$

We first check orthonormality:

$$\mathbf{Q}\mathbf{Q}^\top = \left(\mathbf{I}_d - 2\frac{\mathbf{u}\mathbf{u}^\top}{\|\mathbf{u}\|_2^2}\right)\left(\mathbf{I}_d - 2\frac{\mathbf{u}\mathbf{u}^\top}{\|\mathbf{u}\|_2^2}\right) = \mathbf{I}_d - 4\frac{\mathbf{u}\mathbf{u}^\top}{\|\mathbf{u}\|_2^2} + 4\frac{\mathbf{u}\mathbf{u}^\top\mathbf{u}\mathbf{u}^\top}{\|\mathbf{u}\|_2^4} = \mathbf{I}_d,$$

where the last equality follows since $\mathbf{u}^\top\mathbf{u} = \|\mathbf{u}\|_2^2$. Next, we evaluate $\mathbf{Q}\mathbf{x}$:

$$\mathbf{Q}\mathbf{x} = \left(\mathbf{I}_d - 2\frac{\mathbf{u}\mathbf{u}^\top}{\|\mathbf{u}\|_2^2}\right)\mathbf{x} = \mathbf{x} - 2\mathbf{u} \cdot \frac{\mathbf{u}^\top\mathbf{x}}{\|\mathbf{u}\|_2^2} = \mathbf{x} - 2\mathbf{u} \cdot \frac{1 - \mathbf{t}^\top\mathbf{x}}{2(1 - \mathbf{t}^\top\mathbf{x})} = \mathbf{x} - \mathbf{u} = \mathbf{t}.$$

$\square$

In Section 3.1 we used the following helper result.

**Lemma 16.** *Let* $\mathbf{A}, \mathbf{B} \in \mathbb{S}_{\succeq \mathbf{0}}^{d \times d}$. *Fix* $p \in [d-1]$ *and let* $\mathbf{V} \in \mathbb{R}^{d \times p}$ *have orthonormal columns spanning the top-p eigenspace of* $\mathbf{B}$, *and let* $\widehat{\mathbf{V}} \in \mathbb{R}^{d \times p}$ *have orthonormal columns spanning the top-p eigenspace of* $\mathbf{A}$. *Define* $\mathsf{gap} := \lambda_p(\mathbf{B}) - \lambda_{p+1}(\mathbf{B}) > 0$, *and assume that* $\|\mathbf{A} - \mathbf{B}\|_{\mathrm{op}} \leq \frac{\mathsf{gap}}{4}$. *Then,*

$$\left\|\mathbf{V}\mathbf{V}^\top - \widehat{\mathbf{V}}\widehat{\mathbf{V}}^\top\right\|_{\mathrm{op}} \leq \frac{2\|\mathbf{A} - \mathbf{B}\|_{\mathrm{op}}}{\mathsf{gap}}.$$

*Proof.* Let $\widehat{\mathbf{V}}_\perp \in \mathbb{R}^{d \times (d-p)}$ be any orthonormal basis for $\mathrm{range}(\widehat{\mathbf{V}})^\perp$. Let $\mathbf{E} := \mathbf{A} - \mathbf{B}$, and set

$$\mu := \frac{\lambda_p(\mathbf{B}) + \lambda_{p+1}(\mathbf{B})}{2}, \qquad \tau := \frac{\mathsf{gap}}{2}.$$

By Weyl's inequality,

$$\lambda_{p+1}(\mathbf{A}) \leq \lambda_{p+1}(\mathbf{B}) + \|\mathbf{E}\|_{\mathrm{op}} = \mu - \tau + \|\mathbf{E}\|_{\mathrm{op}} \leq \mu - \frac{\tau}{2} < \mu,$$

and similarly,

$$\lambda_p(\mathbf{A}) \geq \lambda_p(\mathbf{B}) - \|\mathbf{E}\|_{\mathrm{op}} = \mu + \tau - \|\mathbf{E}\|_{\mathrm{op}} \geq \mu + \frac{\tau}{2} > \mu.$$

Hence $\widehat{\mathbf{V}}_\perp$ spans the eigenspace of $\mathbf{A}$ with eigenvalues $\leq \mu$, while $\mathbf{V}$ spans the eigenspace of $B$ with eigenvalues $\geq \mu + \tau$. Applying Wedin's theorem (Fact 1) with $\mathbf{U} = \widehat{\mathbf{V}}_\perp$ yields

$$\left\|\widehat{\mathbf{V}}_\perp^\top\mathbf{V}\right\|_{\mathrm{op}} \leq \frac{\|\mathbf{A} - \mathbf{B}\|_{\mathrm{op}}}{\tau} = \frac{2\|\mathbf{A} - \mathbf{B}\|_{\mathrm{op}}}{\mathsf{gap}}. \tag{28}$$

We now relate this to $\|\widehat{\mathbf{V}}_\perp^\top\mathbf{V}\|_{\mathrm{op}}$. Let $\mathbf{P} := \mathbf{V}\mathbf{V}^\top$ and $\widehat{\mathbf{P}} := \widehat{\mathbf{V}}\widehat{\mathbf{V}}^\top$. Since $\mathbf{P}^2 = \mathbf{P}$ and $\widehat{\mathbf{P}}^2 = \widehat{\mathbf{P}}$,

$$(\mathbf{P} - \widehat{\mathbf{P}})^2 = \mathbf{P} - \mathbf{P}\widehat{\mathbf{P}} - \widehat{\mathbf{P}}\mathbf{P} + \widehat{\mathbf{P}}.$$

Using $\widehat{\mathbf{P}}_\perp := \mathbf{I} - \widehat{\mathbf{P}}$ and $\mathbf{P}_\perp := \mathbf{I} - \mathbf{P}$, we can rewrite this as

$$(\mathbf{P} - \widehat{\mathbf{P}})^2 = \mathbf{P}(\mathbf{I} - \widehat{\mathbf{P}})\mathbf{P} + (\mathbf{I} - \mathbf{P})\widehat{\mathbf{P}}(\mathbf{I} - \mathbf{P}) = \mathbf{P}\widehat{\mathbf{P}}_\perp\mathbf{P} + \mathbf{P}_\perp\widehat{\mathbf{P}}\mathbf{P}_\perp.$$

These two terms act on orthogonal subspaces $\text{range}(\mathbf{P})$ and $\text{range}(\mathbf{P}_\perp)$, hence

$$\left\|(\mathbf{P} - \widehat{\mathbf{P}})^2\right\|_{\text{op}} = \max\left\{\left\|\mathbf{P}\,\widehat{\mathbf{P}}_\perp\,\mathbf{P}\right\|_{\text{op}}, \left\|\mathbf{P}_\perp\,\widehat{\mathbf{P}}\,\mathbf{P}_\perp\right\|_{\text{op}}\right\}.$$

Moreover,

$$\left\|\mathbf{P}\,\widehat{\mathbf{P}}_\perp\,\mathbf{P}\right\|_{\text{op}} = \left\|(\widehat{\mathbf{P}}_\perp\mathbf{P})^\top(\widehat{\mathbf{P}}_\perp\mathbf{P})\right\|_{\text{op}} = \left\|\widehat{\mathbf{P}}_\perp\mathbf{P}\right\|_{\text{op}}^2 = \left\|\widehat{\mathbf{V}}_\perp^\top\mathbf{V}\right\|_{\text{op}}^2,$$

and similarly $\|\mathbf{P}_\perp\,\widehat{\mathbf{P}}\,\mathbf{P}_\perp\|_{\text{op}} = \|\widehat{\mathbf{V}}_\perp^\top\mathbf{V}\|_{\text{op}}^2$. Therefore,

$$\left\|\mathbf{P} - \widehat{\mathbf{P}}\right\|_{\text{op}} = \sqrt{\left\|(\mathbf{P} - \widehat{\mathbf{P}})^2\right\|_{\text{op}}} = \left\|\widehat{\mathbf{V}}_\perp^\top\mathbf{V}\right\|_{\text{op}}.$$

Combining with the Wedin bound in (28),

$$\left\|\mathbf{P} - \widehat{\mathbf{P}}\right\|_{\text{op}} = \left\|\widehat{\mathbf{V}}_\perp^\top\mathbf{V}\right\|_{\text{op}} \leq \frac{\|\mathbf{A} - \mathbf{B}\|_{\text{op}}}{\tau} = \frac{2\,\|\mathbf{A} - \mathbf{B}\|_{\text{op}}}{\text{gap}}.$$

This completes the proof. $\qquad\qquad\square$

## E. Performance of semidefinite programming

In this section, we rederive several short results on the performance of SDP-based algorithms for solving sparse PCA under Model 3. We follow the presentation of (Vu et al., 2013). The following observation motivates the SDP formulations that we consider.

**Fact 4.** *The set* $\{\mathbf{U}\mathbf{U}^\top \mid \mathbf{U} \in \mathbb{R}^{d\times k}, \mathbf{U}^\top\mathbf{U} = \mathbf{I}_k, \bigcup_{i\in[k]} \text{supp}(\mathbf{U}_{:i}) \leq s\}$ *is contained in the convex set*

$$\mathcal{X} := \left\{\mathbf{X} \in k\Delta^{d\times d} \mid \|\mathbf{X}\|_1 \leq sk\right\}, \tag{29}$$

*where* $\Delta^{d\times d} := \{\mathbf{X} \in \mathbb{S}_{\succeq\mathbf{0}}^{d\times d} \mid \text{Tr}(\mathbf{X}) = 1\}$, *and* $\|\mathbf{X}\|_1$ *is applied entrywise.*

Now consider the following SDPs given samples $\{\mathbf{x}_i\}_{i\in[n]}$ from Model 3: the constrained variant

$$\max_{\mathbf{X}\in\mathcal{X}} \left\langle\widehat{\boldsymbol{\Sigma}}, \mathbf{X}\right\rangle \tag{30}$$

and its "unconstrained" counterpart for some $\lambda > 0$:

$$\max_{\mathbf{X}\in\Delta^{d\times d}} \left\langle\widehat{\boldsymbol{\Sigma}}, \mathbf{X}\right\rangle - \lambda\,\|\mathbf{X}\|_1, \tag{31}$$

where we let $\widehat{\boldsymbol{\Sigma}} := \frac{1}{n}\sum_{i\in[n]} \mathbf{x}_i\mathbf{x}_i^\top$ in both (30) and (31). The main fact helpful in analyzing the quality of (30), (31) is a strong convexity bound on the objective $\mathbf{X} \to \langle\boldsymbol{\Sigma}, \mathbf{X}\rangle$.

**Lemma 17.** *In the setting of Model 3, we have*

$$\left\langle\boldsymbol{\Sigma}, \mathbf{V}\mathbf{V}^\top - \mathbf{X}\right\rangle \geq \frac{\gamma\lambda_k(\boldsymbol{\Sigma})}{2}\left\|\mathbf{V}\mathbf{V}^\top - \mathbf{X}\right\|_{\text{F}}^2 \text{ for all } \mathbf{X} \in \mathcal{X}.$$

*Proof.* Throughout the proof, let $\boldsymbol{\Sigma}$ have eigenvalues $\lambda_1 \geq \ldots \geq \lambda_d$ with corresponding eigenvectors $\mathbf{v}_1, \ldots, \mathbf{v}_d$, let the eigendecomposition of $\boldsymbol{\Sigma}$ be $\mathbf{V}\boldsymbol{\Lambda}\mathbf{V}^\top + \mathbf{V}_\perp\boldsymbol{\Lambda}_\perp\mathbf{V}_\perp^\top$ where $\mathbf{V}$ is as in Model 3, and let $\rho_i := \langle\mathbf{X}, \mathbf{v}_i\mathbf{v}_i^\top\rangle$ for all $i \in [d]$. Observe that

$$\left\|\mathbf{V}\mathbf{V}^\top - \mathbf{X}\right\|_{\text{F}}^2 = 2k - 2\left\langle\mathbf{V}\mathbf{V}^\top, \mathbf{X}\right\rangle = 2k - 2\sum_{i\in[k]}\rho_i = 2\sum_{i\in[d]\setminus[k]}\rho_i. \tag{32}$$

Thus our goal is to upper bound $M := \sum_{i \in [d] \setminus [k]} \rho_i$. Next, note that

$$
\begin{aligned}
\left\langle \boldsymbol{\Sigma}, \mathbf{V}\mathbf{V}^\top - \mathbf{X} \right\rangle &= \left\langle \mathbf{V}\boldsymbol{\Lambda}\mathbf{V}^\top, \mathbf{V}\mathbf{V}^\top - \mathbf{X} \right\rangle + \left\langle \mathbf{V}_\perp \boldsymbol{\Lambda}_\perp \mathbf{V}_\perp^\top, \mathbf{V}\mathbf{V}^\top - \mathbf{X} \right\rangle \\
&= \mathrm{Tr}(\boldsymbol{\Lambda}) - \sum_{i \in [k]} \rho_i \lambda_i - \left\langle \boldsymbol{\Lambda}_\perp, \mathbf{V}_\perp^\top \mathbf{X} \mathbf{V}_\perp \right\rangle \\
&= \sum_{i \in [k]} (1 - \rho_i)\lambda_i - \sum_{i \in [d] \setminus [k]} \rho_i \lambda_i \\
&\geq \lambda_k \sum_{i \in [k]} (1 - \rho_i) - \lambda_{k+1} \sum_{i \in [d] \setminus [k]} \rho_i = (\lambda_k - \lambda_{k+1}) \, M.
\end{aligned}
$$

The conclusion follows from our assumption that $\lambda_k - \lambda_{k+1} \geq \gamma \lambda_k$. $\qquad\square$

We next analyze (30) and (31), respectively, by appealing to Lemma 17.

**Proposition 2.** *Let $\delta \in (0,1)$, $\Delta \in (0,1)$, and under Model 3, assume that*

$$
n = \Omega\left( \left( \frac{\sigma^2}{\lambda_k(\boldsymbol{\Sigma})} \right)^2 \frac{s^2 k^2}{\gamma^2 \Delta^2} \log\left( \frac{d}{\delta} \right) \right),
$$

*for an appropriate constant. Let $\mathbf{X}$ be the maximizing argument to (30), following the notation (29). Then with probability $\geq 1 - \delta$, $\left\langle \mathbf{X}, \mathbf{V}\mathbf{V}^\top \right\rangle \geq k - \Delta$.*

*Proof.* First, Fact 2 shows that with probability $\geq 1 - \delta$, for our stated bound on $n$,

$$
\left\| \boldsymbol{\Sigma} - \widehat{\boldsymbol{\Sigma}} \right\|_\infty \leq \frac{\gamma \Delta \lambda_k(\boldsymbol{\Sigma})}{sk}.
$$

Conditioned on this event, we have from Lemma 17, and optimality of $\widehat{\mathbf{X}}$:

$$
\begin{aligned}
\left\| \mathbf{V}\mathbf{V}^\top - \mathbf{X} \right\|_{\mathrm{F}}^2 &\leq \frac{2}{\gamma \lambda_k(\boldsymbol{\Sigma})} \left\langle \boldsymbol{\Sigma}, \mathbf{V}\mathbf{V}^\top - \mathbf{X} \right\rangle \\
&\leq \frac{2}{\gamma \lambda_k(\boldsymbol{\Sigma})} \left\langle \boldsymbol{\Sigma} - \widehat{\boldsymbol{\Sigma}}, \mathbf{V}\mathbf{V}^\top - \mathbf{X} \right\rangle \\
&\leq \frac{2sk}{\gamma \lambda_k(\boldsymbol{\Sigma})} \left\| \boldsymbol{\Sigma} - \widehat{\boldsymbol{\Sigma}} \right\|_\infty \leq 2\Delta,
\end{aligned}
$$

and the conclusion follows by applying the equivalent form (32). $\qquad\square$

We are able to obtain somewhat sharper dependences on $k$ and $\Delta$ by instead using (31).

**Proposition 3.** *Let $\delta \in (0,1)$, $\Delta \in (0,1)$, and under Model 3, assume that*

$$
n = \Omega\left( \left( \frac{\sigma^2}{\lambda_k(\boldsymbol{\Sigma})} \right)^2 \frac{s^2}{\gamma^2 \Delta} \log\left( \frac{d}{\delta} \right) \right),
$$

*for an appropriate constant. Let $\mathbf{X}$ be the maximizing argument to (31), following the notation (29). Then for an appropriate $\lambda$ in (31), with probability $\geq 1 - \delta$, $\left\langle \mathbf{X}, \mathbf{V}\mathbf{V}^\top \right\rangle \geq k - \Delta$.*

*Proof.* Condition on the following event throughout the proof:

$$
\left\| \boldsymbol{\Sigma} - \widehat{\boldsymbol{\Sigma}} \right\|_\infty \leq \frac{\gamma \sqrt{\Delta} \lambda_k(\boldsymbol{\Sigma})}{\sqrt{8} s} =: \lambda,
$$

giving the failure probability for our choice of $n$. Then,

$$0 \le \left\langle \widehat{\mathbf{\Sigma}}, \mathbf{X} - \mathbf{V}\mathbf{V}^\top \right\rangle - \lambda \left( \|\mathbf{X}\|_1 - \|\mathbf{V}\mathbf{V}^\top\|_1 \right)$$

$$= \left\langle \widehat{\mathbf{\Sigma}} - \mathbf{\Sigma}, \mathbf{X} - \mathbf{V}\mathbf{V}^\top \right\rangle - \lambda \left( \|\mathbf{X}\|_1 - \|\mathbf{V}\mathbf{V}^\top\|_1 \right) + \left\langle \mathbf{\Sigma}, \mathbf{X} - \mathbf{V}\mathbf{V}^\top \right\rangle$$

$$\le \lambda \left( \|\mathbf{V}\mathbf{V}^\top - \mathbf{X}\|_1 + \|\mathbf{V}\mathbf{V}^\top\|_1 - \|\mathbf{X}\|_1 \right) - \frac{\gamma \lambda_k(\mathbf{\Sigma})}{2} \|\mathbf{V}\mathbf{V}^\top - \mathbf{X}\|_F^2 .$$

Further, letting $S := \bigcup_{i \in [k]} \mathrm{supp}(\mathbf{v}_i)$ as in Model 3, and using cancellations off the support of $\mathbf{V}\mathbf{V}^\top$,

$$\|\mathbf{V}\mathbf{V}^\top - \mathbf{X}\|_1 + \|\mathbf{V}\mathbf{V}^\top\|_1 - \|\mathbf{X}\|_1 = \left\| [\mathbf{V}\mathbf{V}^\top - \mathbf{X}]_{S \times S} \right\|_1 + \|\mathbf{V}\mathbf{V}^\top\|_1 - \|\mathbf{X}_{S \times S}\|_1$$

$$\le 2 \left\| [\mathbf{V}\mathbf{V}^\top - \mathbf{X}]_{S \times S} \right\|_1 \le 2s \|\mathbf{V}\mathbf{V}^\top - \mathbf{X}\|_F .$$

The desired $\|\mathbf{V}\mathbf{V}^\top - \mathbf{X}\|_F \le \sqrt{2\Delta}$ follows upon plugging in our choice of $\lambda$, and rearranging

$$\frac{\gamma \lambda_k(\mathbf{\Sigma})}{2} \|\mathbf{V}\mathbf{V}^\top - \mathbf{X}\|_F^2 \le 2s\lambda \|\mathbf{V}\mathbf{V}^\top - \mathbf{X}\|_F .$$

$\square$

## F. Additional Experiments

In this appendix, we provide several additional empirical evaluations of our method.

**Scaling ablations.** We study how many samples are needed for the RTPM to recover the top sparse component on both a standard spiked model and the structured non-spiked construction in Lemma 13. In both cases we work in ambient dimension $d = 2500$, run RTPM, set truncation level $r = 10s$, and $T = 100$ iterations, and define $n_{\mathrm{scale}} = \min\{n$ in the grid : $\sin^2 \angle(\hat{\mathbf{v}}, \mathbf{v}) \le \Delta\}$. For the spiked model we draw $\mathbf{x}_i \sim \mathcal{N}(\mathbf{0}, \mathbf{\Sigma})$ with $\mathbf{\Sigma} := (1 - \gamma)\mathbf{I}_d + \gamma \mathbf{v}\mathbf{v}^\top$ where $\mathbf{v}$ has uniform entries $s^{-\frac{1}{2}}$ on its support; for our other experiment we use the worst-case block construction in Lemma 13 (with the second eigenvalue $\lambda_2 = 1 - \gamma$) embedded into $d = 2500$ by padding the remaining coordinates with isotropic variance $1 - \gamma$. Across all sweeps, the empirical behavior is consistent and monotone: $n_{\mathrm{scale}}$ increases with $s$, decreases as $\gamma$ grows, and decreases as the target error $\Delta$ is relaxed.

We also fit degree-2 and 3 polynomials to the curves in Figures 3a and 3b, and power functions with negative exponents 1 and 2 for Figures 3c, 3d, 3e, 3f to understand the variation of the sample size with these parameters. For dependence on $s$, both 2 and 3 degree polynomial-fit curves capture most of the variation. The plots also suggest that the dependence of the error of our algorithm on $\gamma$ and $\Delta$ is possibly more moderate than that suggested by the theoretical results.

**Real data.** We experiment on a standard text sparse PCA setup in (Zhang and Ghaoui, 2011) using the publicly available NYTimes bag-of-words dataset from the UCI "Bag of Words" collection (Newman, 2008). Each document is represented as a sparse word count vector. We restrict to the first $n = 10000$ documents, and we select a vocabulary of size $d = 20000$. We apply a stabilizing transform $\log(1 + c)$ to word-document counts and center the data by subtracting the empirical feature mean. We compute $k = 4$ sparse principal components via a deflation-style routine (Algorithm 2) coupled with the RTPM algorithm (Algorithm 2) with $r = 50$ and $T = 50$. The output is a sparse vector whose largest-magnitude entries correspond to an interpretable set of words (see Figure 5).

The resulting sparse components cleanly separate semantic themes in the corpus. From the top-10 words by absolute value of entries, the leading components align with (i) sports and game coverage (PC 1), (ii) US politics and elections (PC 2), (iii) markets, business, and finance (PC 3), and (iv) web, publishing, and metadata terms (PC 4). The "union-support" in Figure 4 shows that most energy of each component is concentrated in its top ranks, consistent with sparsity. Our findings in this application are very *interpretable*: each component is supported on a small set of words, making the latent topic structure easy to understand compared to dense PCA. This aligns with the results in (Zhang and Ghaoui, 2011).

We note that Lemma 9 is a worst-case construction showing that deflation is not provably closed under Model 3: after projecting out a sparse vector that is arbitrarily well aligned with $\mathbf{v}_1$, the residual top eigenvector can become dense. In the NYT data, vocabulary naturally clusters into well-separated semantic topics. As a result, after deflation, the support

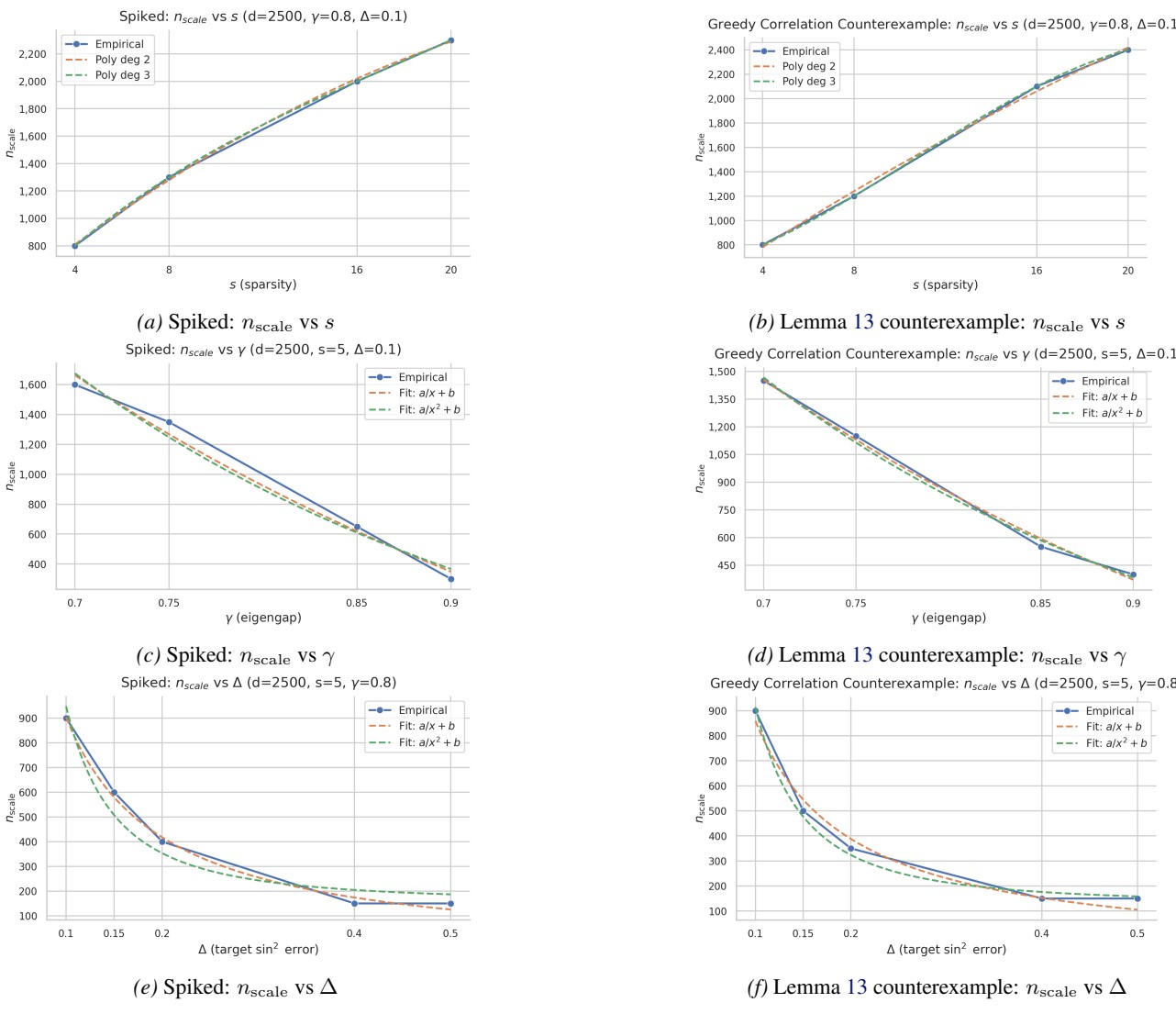

*(a)* Spiked: $n_{\text{scale}}$ vs $s$

*(b)* Lemma 13 counterexample: $n_{\text{scale}}$ vs $s$

*(c)* Spiked: $n_{\text{scale}}$ vs $\gamma$

*(d)* Lemma 13 counterexample: $n_{\text{scale}}$ vs $\gamma$

*(e)* Spiked: $n_{\text{scale}}$ vs $\Delta$

*(f)* Lemma 13 counterexample: $n_{\text{scale}}$ vs $\Delta$

*Figure 3.* Scaling-law experiments for RTPM. Left column: spiked covariance model. Right column: Lemma 13 counterexample. Rows correspond to varying $s$, $\gamma$, and $\Delta$, respectively.

sets of distinct topics have limited overlap, which does not follow the worst-case construction. Lemma 9 is a theoretical barrier showing that provable guarantees for deflation require new ideas and does not predict that deflation will always fail empirically.

Since ground-truth eigenvectors are unavailable on real data, we report the following metrics:

- *Explained Variance Ratio (EVR):* $u_i^\top \widehat{\Sigma} u_i / \text{Tr}(\widehat{\Sigma})$, measuring variance captured per component.

- *Normalized Pointwise Mutual Information (NPMI) topic coherence:* for each PC, we take the top-10 words by loading magnitude and compute the average pairwise NPMI using document co-occurrence, where

$$\text{NPMI}(w_i, w_j) = \frac{\log P(w_i, w_j) - \log P(w_i) - \log P(w_j)}{-\log P(w_i, w_j)},$$

and probabilities are estimated from document-level word presence. NPMI $\in [-1, 1]$ serves as a standard proxy for human-judged topic interpretability.

- *Effective sparsity (NNZ / PC):* number of non-zero entries per PC.

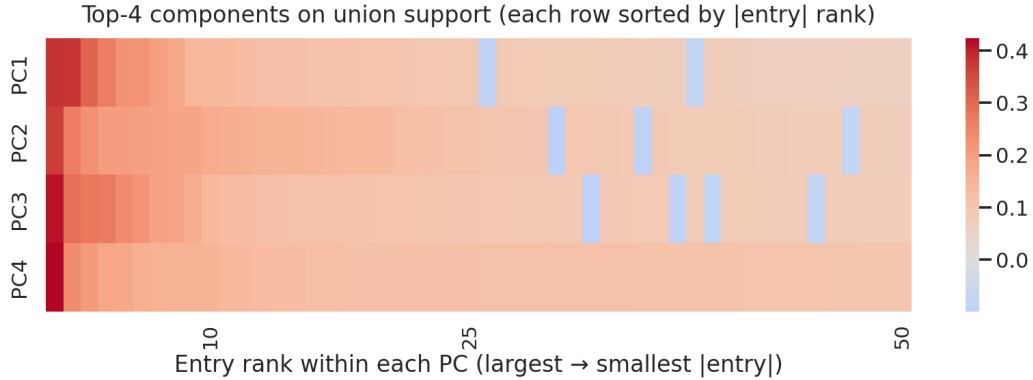

*Figure 4.* Top 4 components restricted to the union support, with each row sorted by rank of entries.

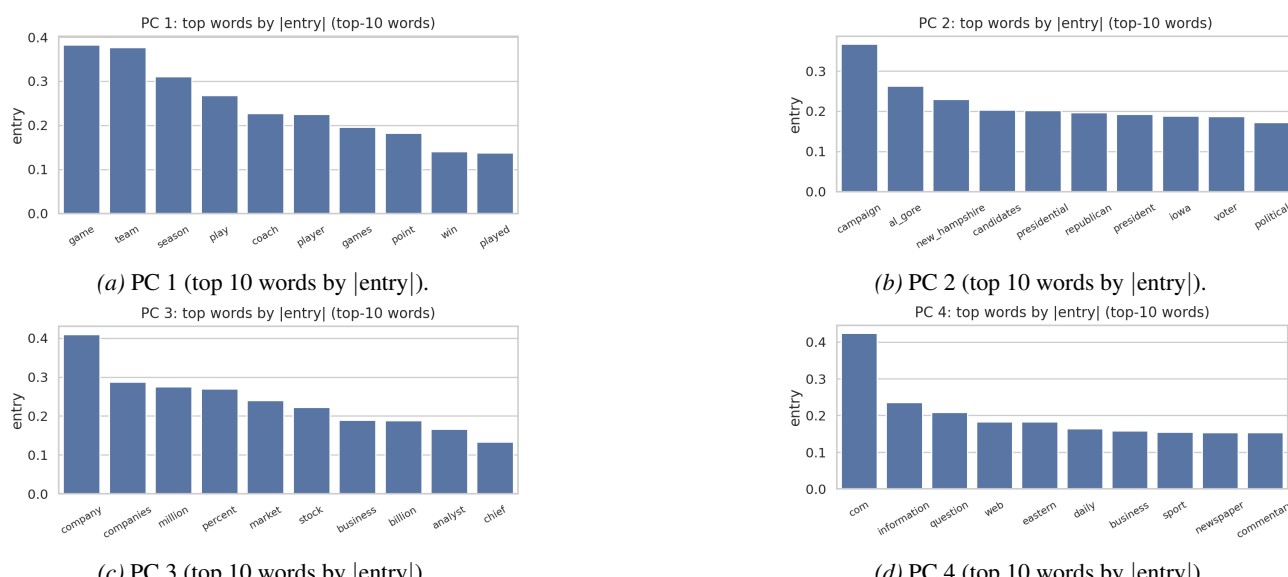

*(a)* PC 1 (top 10 words by |entry|).      *(b)* PC 2 (top 10 words by |entry|).

*(c)* PC 3 (top 10 words by |entry|).      *(d)* PC 4 (top 10 words by |entry|).

*Figure 5.* Top 10 words associated with each sparse principal component.

We compare against DiagThresh (Alg. 3), CovThresh (Alg. 4, with $\tau$ tuned to the data-dependent covariance scale), and GreedyCorr (Alg. 5, 200 seeds). All methods use $s = 50$, $k = 5$. SDP is omitted as it is infeasible at $d = 20,000$.

| Method | Total EVR (%) | Avg NPMI | NNZ / PC |
|---|---|---|---|
| **RTPM (Ours)** | **3.0** | **0.325** | **50** |
| DiagThresh | 2.28 | 0.130 | 50 |
| CovThresh | 2.29 | 0.283 | 31–197 |
| GreedyCorr | 3.69 | 0.308 | 50 |

RTPM achieves the best EVR and NPMI. DiagThresh performs substantially worse, showing that marginal-variance support selection is inadequate beyond the spiked identity setting. CovThresh attains moderate coherence but has unstable sparsity and requires manual threshold tuning. GreedyCorr is competitive on this dataset, which does not contradict our theory; our lower bound only shows failure on adversarial instances. RTPM is the only method here that combines strong empirical performance with provable guarantees under general covariances.

