# OpenReview forum: "Combinatorial Sparse PCA Beyond the Spiked Identity Model"
_ICML.cc/2026/Conference — ICML 2026 regular_

### Official Review · Reviewer_QrNN · 2026-03-11

**Soundness:** 3
**Presentation:** 3
**Significance:** 3
**Originality:** 3
**Overall Recommendation:** 5
**Confidence:** 5

**Summary:**

This paper investigates the problem of Sparse Principal Component Analysis in high-dimensional settings. The authors focus on the theoretical and algorithmic challenges that arise when moving beyond the conventional Spiked Identity Model. To address this gap, the authors propose a method titled the Restarted Truncated Power Method and establish its global convergence guarantees under the general Model 2 and Model 3. A key theoretical contribution is the analysis of deflation failure in sparse subspace recovery. Specifically, the authors use Lemma 9 to construct a counterexample showing that standard deflation can lead to a loss of sparsity in the residual matrix principal components. The performance of the proposed method is validated on both synthetic data and the New York Times real-world dataset.

**Compliance With Llm Reviewing Policy:**

Affirmed.

**Key Questions For Authors:**

1. Theorem 1 establishes a sample complexity of $s^3 \log d$. Compared to the $s^2 \log d$ complexity achieved by methods based on Semidefinite Programming, there is an additional factor of $s$. The authors should clarify if this gap is due to deficiencies in the proof techniques, such as the correlation analysis in Lemma 6, or if it represents an inherent statistical limitation of the RTPM algorithm when moving beyond the spiked identity model.

2. Lemma 9 provides a crucial counterexample to the failure of deflation under Model 3, yet experiments on the New York Times dataset appear to have been successful. Please provide statistical intuition to explain why real-world data did not trigger the worst-case scenario described in the lemma. Could this success be related to inconsistencies or structural overlap in the support sets?

3. Is the deflation failure revealed in Lemma 9 a worst-case existence result limited to specific constructions, or is it a more general phenomenon that occurs in practice? The authors should clarify the practical prevalence of this failure mechanism.

4. For the four principal components extracted from real-world data, the authors should provide quantitative metrics rather than relying solely on qualitative descriptions of keywords.

5. Since the theory requires the truncation parameter $r$ to be at least at the level of $s^2$, but the sparsity $s$ is often unknown in practical applications, the authors should supplement the paper with a sensitivity analysis regarding the choice of the parameter $r$.

6. The current experimental evaluation should be strengthened through a more comprehensive comparison with baseline methods. While some comparisons exist in the simulations, the actual data analysis only shows the results of the proposed algorithm.

7. The parameter $\gamma$ is introduced when Model 1 and Model 2 are first presented but it does not seem to appear in the actual structural specifications of the models. The authors should clarify whether $\gamma$ is part of the model specification and explicitly incorporate it into the model definitions.

8. In the decomposition of the covariance matrix in the first two lines of Equation 5, the authors utilize a term $\Lambda_p$ which is not predefined. While the appendix defines this symbol as the eigenspace, the main text uses $V_p$ for the same purpose. The authors should standardize the notation throughout the manuscript to avoid confusion between eigenspaces and eigenvalue matrices.

9. The clarity of the writing could be improved. In its current form, the structure and presentation do not always clearly highlight the main ideas. A clearer exposition would improve the readability of the paper.

**Limitations:**

yes

**Strengths And Weaknesses:**

Strengths.
The paper studies an important sparse PCA problem beyond the spiked identity model and makes a meaningful conceptual contribution by identifying failure modes of existing combinatorial/deflation-based methods, proposing RTPM, and providing theoretical guarantees under a more general covariance model. The paper is also empirically useful, with synthetic and real-data experiments showing strong interpretability and competitive performance.

Weaknesses.
The main weakness is that the current theory still has a sample complexity gap relative to stronger SDP-based results, and the practical choice of the truncation parameter 𝑟 is not fully justified. In addition, the real-data evaluation is mostly qualitative, and the paper would benefit from clearer notation, stronger baseline comparisons in real-data experiments, and an explicit code release for reproducibility.

---

> ### Author Rebuttal · Authors · 2026-03-31
>
> We thank the reviewer for their positive assessment, and for their helpful suggestions. We will address all notational and presentation issues in the revision.
>
> **Improved sample complexity.** We are excited to share that since submission, we have improved the sample complexity in Theorem 2 from $n = \Omega(s^3 \log d)$ to $\Omega(s^2 \log d)$, matching the optimal sample complexity at the computational-statistical threshold. See our response to reviewer mDvb for more details.
>
> **Quantitative metrics and baseline comparisons.** We have expanded our NYTimes evaluation ($n=10,000$, $d=20,000$) with quantitative metrics and comparisons against all combinatorial baselines from our paper. Since ground-truth eigenvectors are unavailable on real data, we report the following metrics:
>
> - *Explained Variance Ratio (EVR)*: $u_i^\top \hat{\Sigma} u_i / \mathrm{tr}(\hat{\Sigma})$, measuring variance captured per component.
>
> - *Normalized Pointwise Mutual Information (NPMI) topic coherence* [A]: for each PC, we take the top-10 words by loading magnitude and compute the average pairwise NPMI using document co-occurrence, where $\text{NPMI}(w_i, w_j) = \frac{\log P(w_i, w_j) - \log P(w_i) - \log P(w_j)}{-\log P(w_i, w_j)}$ and probabilities are estimated from document-level word presence. NPMI ($\in [-1,1]$) serves as a standard proxy for human-judged topic interpretability [A,B,C].
>
> - *Effective sparsity (NNZ / PC)*: number of non-zero entries per PC.
>
> We compare against DiagThresh (Alg. 3), CovThresh (Alg. 4, with $\tau$ tuned to the data-dependent covariance scale), and GreedyCorr (Alg. 5, 200 seeds). All methods use $s=50$, $k=5$. SDP is omitted as it is infeasible at $d=20,000$.
>
> | Method | Total EVR (%) | Avg NPMI | NNZ / PC |
> |---|---|---|---|
> | **RTPM (Ours)** | **3.80** | **0.325** | **50** |
> | DiagThresh | 2.28 | 0.130 | 50 |
> | CovThresh | 2.29 | 0.283 | 31–197 |
> | GreedyCorr | 3.69 | 0.308 | 50 |
>
> RTPM achieves the best EVR and NPMI. DiagThresh performs substantially worse, showing that marginal-variance support selection is inadequate beyond the spiked identity setting. CovThresh attains moderate coherence but has unstable sparsity and requires manual threshold tuning. GreedyCorr is competitive on this dataset, which does not contradict our theory; our lower bound only shows failure on adversarial instances. RTPM is the only method here that combines strong empirical performance with provable guarantees under general covariances.
>
> **Sensitivity of truncation parameter $r$.** We ran RTPM on the Lemma 4 counterexample ($s=4$, $\gamma=0.2$, $d =600$) with $r$ ranging from $s/4$ to $8s$:
>
> | $r$ | $r/s$ | $\cos^2(u, v)$, spiked model | $\cos^2(u, v)$, Lemma 13 model |
> |---|---|---|---|
> | 1 | 0.25 | 0.250 | 0.250 |
> | 2 | 0.5 | 0.491 | 0.161 |
> | 4 ($=s$) | 1 | 0.986 | **0.9999** |
> | 8 | 2 | 0.949 | 0.996 |
> | 16 | 4 | 0.872 | 0.996 |
> | 32 | 8 | 0.774 | 0.996 |
>
> With large $n$, $r = s$ already achieves near-perfect recovery, and all $r \geq s$ yield $\cos^2 > 0.99$ for non-spiked model. Only $r < s$ fails, as expected, since the iterates cannot retain the full support. This confirms that the theoretical requirement $r = \Omega(s^2)$ is conservative; in practice, $r$ on the order of $s$ suffices, and the empirical performance is robust to the choice of $r$ as long as $r \geq s$.
>
> **Lemma 9 and empirical deflation.** Lemma 9 is a worst-case construction showing that deflation is not provably closed under Model 3: after projecting out a sparse vector that is arbitrarily well aligned with $v_1$, the residual top eigenvector can become dense. In the NYT data, vocabulary naturally clusters into well-separated semantic topics. As a result, after deflation, the support sets of distinct topics have limited overlap, which does not follow the worst-case construction. Lemma 9 is a theoretical barrier showing that provable guarantees for deflation require new ideas and does not predict that deflation will always fail empirically.
>
> **$\gamma$ in model definition.** We apologize for the confusion. In Models 1 and 2, we fix $\gamma = 0.1$ for simplicity, to focus on the dependence on $s$ and $d$. We use general $\gamma$ in Model 3 and Thm 2 to show our algorithm handles arbitrary eigengap. We will clarify this in the revision.
>
> **Notation ($\Lambda_p$ in Equation 5).** $\Lambda_p$ is the diagonal matrix of the top-$p$ eigenvalues (not the eigenspace, which is $V_p$). We will add an explicit definition before Eq. 5 and ensure consistency between the main text and appendix.
>
> **Code release.** We provide our code in the supplementary material included with our current submission.
>
> [A] Lau, Newman & Baldwin. *Machine Reading Tea Leaves: Automatically Evaluating Topic Coherence and Topic Model Quality.* EACL 2014.
>
> [B] Newman, Lau, Grieser & Baldwin. *Automatic Evaluation of Topic Coherence.* NAACL 2010.
>
> [C] Röder, Both & Hinneburg. *Exploring the Space of Topic Coherence Measures.* WSDM 2015.

---

> > ### Author Rebuttal · Reviewer_QrNN · 2026-04-03
> >
> > No further comments have been provided.

---

### Official Review · Reviewer_mDvb · 2026-03-11

**Soundness:** 3
**Presentation:** 3
**Significance:** 3
**Originality:** 3
**Overall Recommendation:** 4
**Confidence:** 4

**Summary:**

The paper addresses the sparse principal component analysis (PCA) problem under a general covariance model (Model 2 and Model 3) that relies only on a sparse leading eigenvector and an eigengap, relaxing the stringent spiked identity model assumption (Model 1). The authors demonstrate that standard combinatorial heuristics fail under this generalized model at information-theoretically optimal sample complexities. To bridge the gap between fast combinatorial methods and robust but computationally expensive semidefinite programming (SDP) methods, they propose a restarted truncated power method (RTPM). They prove this algorithm converges globally without requiring local warm starts, achieving recovery using $\text{poly}(s, \log d)$ samples and running in $d^2\text{poly}(s, \log(d))$ time. Additionally, the paper establishes a theoretical barrier, showing that standard deflation-based reductions for sparse $k$-PCA can fail catastrophically under the general model.

**Compliance With Llm Reviewing Policy:**

Affirmed.

**Final Justification:**

This paper makes a definitive theoretical contribution to sparse k-PCA by presenting the first polynomial-time algorithm (RTPM) that achieves global convergence under a generalized covariance model, avoiding the local initialization needed in prior work. Its most important results consist of two key contributions: rigorous counterexamples that expose vulnerabilities in common heuristics at the conjectured optimal sample complexity, and a theoretical barrier against naive deflation. Together, these insights substantially advance the understanding of the problem's computational-statistical trade-offs.

My main concern was the severe suboptimality of the sample complexity and the dense truncation parameter, which would limit practical utility. The authors have convincingly addressed this concern: not only have they achieved the optimal Ω(s^2 log d) sample complexity, which constitutes a significant technical improvement, but they have also provided empirical evidence indicating that the dependence on other parameters is milder in practice. They have also committed to improve presentation, such as moving key experiments to the main text.

Although the current bounds still show large polynomial dependencies on the eigengap and precision, these do not undermine the paper’s core theoretical contributions: closing a known computational gap and establishing important negative results. With the promised revisions, I recommend Weak Accept (4). This is a clear and meaningful advance in the theory of sparse PCA.

**Key Questions For Authors:**

1. You present Lemma 9 as a strong barrier against deflation-based $k$-sparse PCA under Model 3. Yet, your primary real-world experiment (NYTimes, Appendix F) utilizes exactly this deflation strategy (Algorithm 2) to successfully extract topics. How do you reconcile this theoretical impossibility with your empirical success? Is the NYTimes dataset simply closer to the spiked identity model (Model 1)?
2. Theorem 2 requires $n \propto 1/(\Delta^{10}\gamma^{10})$. Is this exorbitant polynomial dependence a fundamental bottleneck of the restarted truncated power method, or is it merely a consequence of loose analytical bounds (e.g., compounding union bounds)?
3. Your elegant counterexamples (Lemmas 10, 12, 13) prove failure at the optimal sample complexity $n = \Omega(s^2 \log d)$. Do these heuristics remain completely broken at larger sample sizes (e.g., $n = \Omega(s^3 \log d)$), or does excess data eventually rescue these standard algorithms under Model 2?
4. Restarting from all $d$ standard basis vectors guarantees global convergence but feels like a computationally brute-force workaround. Is there a theoretical path to initializing from a much smaller candidate set (e.g., selected via marginal variances) while still preserving the global guarantees under Model 2?

**Limitations:**

The paper's Impact Statement does not adequately address limitations. The authors should explicitly discuss: (1) the practical implications of the $\gamma^{-10}$ and $\Delta^{-10}$ sample complexity dependence for datasets with small eigengap; (2) the unresolved theoretical contradiction between Lemma 9 (deflation fails under Model 2) and its practical efficacy in the NYTimes experiment, perhaps by discussing whether real-world data implicitly satisfies the stricter Model 1 conditions; and (3) sensitivity to violations of the sub-Gaussian assumption.

**Strengths And Weaknesses:**

*Strengths:*

Soundness:
- The theoretical analysis establishing global convergence for the RTPM algorithm under generalized covariance models is mathematically rigorous and successfully circumvents the need for the local initialization assumed in prior work.
- The mathematical counterexamples (Lemmas 10, 12, 13) are meticulously constructed, proving the vulnerability of existing heuristics (diagonal/covariance thresholding and greedy correlation) at the conjectured minimal sample complexity $n = \Omega(s^2 \log d)$.

Presentation:
- The motivation via the theoretical computer science semi-random framework is excellent, providing a compelling narrative for why robustness beyond the spiked identity model is necessary.
- The progression from Model 1 (Spiked Identity) to Model 3 (General Sparse $k$-PCA) is logically structured, making the theoretical problem boundaries very clear.

Significance:
- Closing the computational gap between Model 1 and Model 2 by providing a polynomial-time combinatorial algorithm for general sparse PCA addresses a fundamental open problem in high-dimensional statistics.
- The deflation barrier (Lemma 9) is a highly significant negative result that rigorously challenges the prevailing "self-reducibility" assumptions in the sparse $k$-PCA literature.

Originality:
- Applying the semi-random framework (specifically adapting the planted clique "greedy correlation" counterexample) to sparse PCA reveals entirely new vulnerabilities in widely accepted algorithms.
- Developing a theoretical barrier against sparse PCA deflation strategies (Lemma 9) is a creative and highly original contribution.


*Weaknesses:*

Soundness:
- The sample complexity in Theorem 2 is severely suboptimal, not just in its $s^3 \log d$ scaling, but fatally in its dependence on the eigengap $\gamma$ and target error $\Delta$. The requirement of $n \propto 1/(\Delta^{10}\gamma^{10})$ renders the theoretical guarantee practically meaningless for ill-conditioned datasets or reasonable precision requirements.
- The truncation parameter $r = \Omega(s^2 k^2 / (\Delta^2 \gamma^2) + \dots)$ is highly superlinear in $s$. This means the intermediate vectors are extremely dense, which undermines the practical memory and computational benefits expected from a "sparse" algorithm.

Presentation:
 - The paper relies far too heavily on the appendix. Crucial narrative elements, including the explicit counterexamples motivating the paper and the core real-world experiments on the NYTimes dataset, are entirely relegated to the supplementary material.
- The informal statement of Theorem 1 does not display the algorithm's heavy dependence on the eigengap $\gamma$ (by quietly fixing $\Delta = \gamma = 1/10$), which misleads readers about the true statistical cost of the algorithm until they reach page 7.

Significance:
- The practical significance of RTPM is severely limited by the exorbitant polynomial dependencies on $\gamma$ and $\Delta$. It is highly questionable whether RTPM can practically replace SDPs in regimes where SDPs are currently utilized.
- There is a critical methodological discrepancy regarding significance: The authors prove that deflation-based sparse PCA fails theoretically under their general model (Lemma 9), yet they rely on exactly this theoretically defunct deflation strategy (Algorithm 2) to generate their real-world NYTimes results in Appendix F.

Originality:
- The core algorithmic primitive—the truncated power method—is not novel and is directly borrowed from Yuan and Zhang (2013).
- Restarting from all $d$ basis vectors constitutes an exhaustive initialization strategy rather than a fundamentally novel algorithmic mechanism; the contribution lies primarily in showing it suffices and analyzing it rigorously.

---

> ### Author Rebuttal · Authors · 2026-03-31
>
> Thank you for your positive comments. We will restructure and incorporate more empirical results into the main body, and add an intuitive overview for Section 3.
>
> **Dependence in $s$.** We highlight that recently we improved the sample complexity in Thm 2 from $n = \Omega(s^3 \log d)$ to $\Omega(s^2 \log d)$, matching the conjectured optimum. This comes from a novel normalized potential and stronger bilinear form concentration, which bypasses the previous operator-norm bottleneck and is a significant technical advance over [YZ13]. We will include this improvement in our final version.
>
> In each iteration of the first phase of our current proof, the power-method update increases the potential $\phi_t$ proportionally, but suffers from sampling error. Bounding the operator norm difference between population and empirical submatrices leads to an $O(\sqrt{r/n})$ error; balancing with the $O(\sqrt{s/r})$ truncation error forced $r \approx s^2$ and $n \gtrsim s^3$.
>
> We devise an improved normalized potential that bypasses these operator norm-based bounds. It allows us to bound the sampling error directly by controlling a suitable quadratic form $|\langle \hat v_t, \Sigma-\widehat\Sigma_t v_1\rangle|$ where $\hat v_t$ is the $t^{th}$ iterate and $v_1$ is the population eigenvector. This relies on the independence of the iterates and the batch empirical covariance, achieved via sample splitting across $t$, and reduces sampling error to $O(1/\sqrt{n})$. Consequently, keeping $r \approx s^2$ to control the truncation error now enables $n = \Omega(s^2 \log d)$.
>
> **Dependence on $\Delta$ \& $\gamma$.**, **Significance.**
>
> We believe that the $(\gamma\Delta)^{-10}$ rate is an artifact of our analysis. It mainly comes from bounding the potential when $\phi_t \in [1/4, 1-\Delta]$ in Lemma 7. When close to $1 - \Delta$, progress slows leading to an additional $\text{poly}(\gamma^{-1}, \Delta^{-1})$ factor. Our improved analysis above achieves a better $(\gamma\Delta)^{-6}$. Our primary focus was to achieve the 1st fast algorithm with $\text{poly}(s)\log(d)$ samples under Model 2, with the optimal dependence on other parameters being a natural next direction.
>
> Empirically, Figure 3 (Appendix F, page 27) suggests that the dependence on $\Delta^{-1} $ and $ \gamma^{-1}$ may be much milder than shown in our analysis, and is on par with the strongest theoretical results. Finally, regardless of our dependence on $\gamma\Delta$, our method offers an immediate practical advantage over SDPs in terms of memory usage, a bottleneck encountered even at moderate dimension in our SDP experiments.
>
> **Deflation on NYtimes data.** Please refer to the response to reviewer QrNN.
>
> **$n$ in counterexamples.** Increasing sample size does not rescue these algorithms: our counterexamples hold for all $n \geq \Omega(s^2 \log d)$, as stated in Lemmas 10, 12, and 13.
>
> **$d$ restarts.** We believe that in the worst case, restart count cannot be drastically improved beyond $\Omega(d)$. The counterexample in Lemma 10 shows that, in Model 2, marginal variances are insufficient for selecting the support.
>
> **Originality.** We respectfully disagree that our contribution is merely a new analysis of an existing method. Truncated power method has been known since 2013, yet achieving global convergence under Model 2 remains open. We highlight a key distinction: retaining the top $r \approx s^2$ coordinates is a novel design crucial for global convergence. This oversampling allows sufficient room to preserve progress during the initial low-correlation regime. The $d$ restarts provide only a weakly-correlated seed, while the modified truncation is crucial in recovery beyond the local setting of [YZ13].
>
> **Truncation parameter.** The dependence of $r$ on $\gamma\Delta$ comes from the same bottleneck discussed above. In our response to Reviewer QrNN, we provide tests that empirically show that the suboptimal rate in Thm 2 is unnecessary. We provide an additional test to show how $\gamma$ trades off $r$ when the empirical error $\Delta$ is of the same level on the model in Lemma 12 with $d = 600$, $s = 4$ and $n=4000$.
> | $\gamma$ | $r$ | $\Delta$ (%) |
> |---|---|---|
> |0.25|7|1.4|
> |0.2|8|2.6|
> |0.15|10 |2.4|
> |0.1|14|1.9|
>
> This also indicates $\gamma$ only roughly scale with $1/r$. Furthermore, even in the worst case, maintaining a dense vector is not the main bottleneck, since the computation remains $O(nd^2)$ and the memory remains $O(nd)$, which is significantly faster than SDPs.
>
> **Informal Thm 1.** The intention for Thm 1 was to highlight the $s$ dependence, and thus we fixed $\gamma = \Delta = 0.1$. Our intention was not to mislead readers, and we will add a remark to explicitly note the dependence is large. We note that the ICML guidelines state that the Impact Statement is for broader societal and ethical impacts.
>
> **Sub-Gaussian assumption.** Please refer to the response to Reviewer Ha7F.

---

> > ### Author Rebuttal · Reviewer_mDvb · 2026-04-01
> >
> > Thanks for your rebuttal. I remain my original score.

---

> > > ### Author Response · Authors · 2026-04-03
> > >
> > > Thank you again for reading the rebuttal carefully. We appreciate your note that the concerns were fully resolved. If there are any remaining technical points, clarifications, or presentation improvements that would be useful for the discussion, we would be very happy to address them.

---

### Official Review · Reviewer_xzLh · 2026-03-12

**Soundness:** 3
**Presentation:** 3
**Significance:** 3
**Originality:** 3
**Overall Recommendation:** 5
**Confidence:** 3

**Summary:**

In this paper, the authors consider the problem of estimating the low-rank, sparse structure in the covariance matrix. Assuming only that the top eigenvector of the covariance matrix is sparse and enough gap between the top two eigenvalues, instead of the usual rank-1 spike assumption, the authors propose a computationally efficient combinatorial algorithm that well-approximates the leading eigenvector of the covariance matrix. The algorithm is based on the modification of the truncated power method. Mathematical analysis on the performance guarantee and the complexity is given.

**Compliance With Llm Reviewing Policy:**

Affirmed.

**Final Justification:**

The authors answered my question adequately. I will maintain my score.

**Key Questions For Authors:**

- It seems that the number of the samples $n$ in some experiments is too large (over the BBP threshold). I wonder if it would possibly affect the comparison of the algorithms.

**Limitations:**

Yes

**Strengths And Weaknesses:**

- The proposed algorithm is highly competitive in terms of the complexity when compared to other known algorithms.
- The assumption on the structure of the covariance matrix is weaker than many other related works.
- The manuscript is well-written and the results are well-presented.

---

> ### Author Rebuttal · Authors · 2026-03-31
>
> We thank the reviewer for their positive comments about our algorithm, presentation and writing.
>
> **On sample sizes in experiments**:
> We would like to clarify that we chose $n$ and $d$ in these experiments to ensure fair comparisons. Particularly, SDP-based methods scale poorly and suffer from major computational bottlenecks at large $d$. Under this dimension constraint, $\mathsf{poly}(s)\log(d)$, which is the required sample complexity for non-trivial convergence of all algorithms, is indeed close to the BBP threshold.
> Furthermore, even when $n$ is above the BBP threshold for dense PCA, sparse PCA remains nontrivial to achieve error below $O(d/n)$, which our experiments demonstrate. The BBP threshold governs when the top dense eigenvector has nontrivial correlation to the sparse population eigenvector. However it does not guarantee whether a given algorithm can reliably recover the sparse top component under minimal structural assumptions. Indeed, our counterexample experiments (Figure 2) show clear separation between methods even at large $n$: DiagThresh and CovThresh fail on the adversarial instances regardless of sample size, while RTPM (our algorithm) succeeds.
>
> **Improved sample complexity**: We would also like to take this opportunity to mention that since submitting our original manuscript, we have further improved the sample complexity in Theorem 2 from $n = \Omega(s^3 \log d)$ to $n = \Omega(s^2 \log d)$, matching the optimal sample complexity at the computational-statistical threshold. The improvement comes from a novel normalized potential function and bilinear form concentration, which bypasses the operator-norm bottleneck in our original analysis. We will include this improved result in our revised manuscript. Please see the response to Reviewer mDvb for further details.

---

> > ### Author Rebuttal · Reviewer_xzLh · 2026-04-01
> >
> > The authors answered my question adequately.

---

### Official Review · Reviewer_Ha7F · 2026-03-12

**Soundness:** 4
**Presentation:** 2
**Significance:** 4
**Originality:** 3
**Overall Recommendation:** 5
**Confidence:** 2

**Summary:**

Principal component analysis is one of the fundamental tools in computer science, widely used across machine learning, data mining, and as a subroutine in numerous algorithms and specific applications. Sparse PCA represents one of the most critical problem classes for circumventing the shortcomings of standard PCA. However, due to its combinatorial optimization problem nature, developing efficient solutions with theoretical guarantees has remained a non-trivial, long-standing challenge. This paper presents a robust solution to this question, offering a new perspective inspired by tools from sub-random models.

**Compliance With Llm Reviewing Policy:**

Affirmed.

**Final Justification:**

This paper is a highly insightful piece of work that resolves one of the long-standing challenges in sparse PCA. Personally, I believe this paper is ready to be presented to the broader machine learning community. My only request to the authors is that they exercise caution regarding whether to include the theoretical results that emerged after the initial submission in the main text of the camera-ready version (it might be appropriate to present major results that emerged after the initial submission at a different time or in a different journal).

**Key Questions For Authors:**

I would like to thank the authors for sharing this excellent paper. I can follow the general flow of Section 3 (the theoretical main result outlined in Theorem 1) and accept its validity in a broad sense. Specifically, I was able to follow the flow up to Lemma 4. On the other hand, I have not been able to keep up with the technical details of Lemmas 5, 6, and 7, and I have only skimmed the proofs provided in the supplementary materials.

I unreservedly agree that these results, if correct, are of extremely high importance and significance. I also get the impression that these arguments seem to possess considerable validity. However, as I am not an expert in this topic, I have not been able to verify the detailed correctness of these arguments (though I have tried my best). I hope there are theorists specializing in this topic among the reviewers and meta-reviewers. ICML is indeed a conference that covers theoretical aspects of machine learning, making it an excellent venue for this topic. However, PCA seems to have aspects that lean more towards pure computer science than recent machine learning trends. If the authors wish to share this paper with the community as a fundamental and important result in computer science, venues like STOC or FOCS might also be strong candidates. Indeed, as the authors note, the prior work [Blasiok+2024] that strongly influenced this paper was presented at FOCS, suggesting these conferences may be more suitable. Given the above, I hope that someone within this peer review process can thoroughly verify the correctness of this paper from a theoretical perspective.

Specifically, I have not gained a clear understanding of the following points. I do not fully understand why the assumption of sub-Gaussian data distribution is necessary for this paper. At first glance, it appears sub-Gaussianity is required for the inequalities in Facts 2 and 3 provided in Appendix A. However, similar rate inequalities are known to hold for a broader class of probability distributions in the form of matrix Chernoff bounds [Kyng+FOCS2018; Gillman, SIAM j. Comp. 1998]. More specifically, I am thinking of inequalities in the form of, for example, Theorem 1.2 in the reference [Kaufman+, SODA2022].
Is the sub-Gaussian assumption truly necessary for the paper's argument? Or can the results be broadly extended to data distributions where the Central Limit Theorem or the Law of Large Numbers hold?

- Kyng, R., & Song, Z. (2018, October). A matrix chernoff bound for strongly rayleigh distributions and spectral sparsifiers from a few random spanning trees. In 2018 IEEE 59th Annual Symposium on Foundations of Computer Science (FOCS) (pp. 373-384). IEEE.

- Gillman, D. (1998). A Chernoff bound for random walks on expander graphs. SIAM Journal on Computing, 27(4), 1203-1220.

- Kaufman, T., Kyng, R., & Soldá, F. (2022). Scalar and matrix chernoff bounds from
L\infty-independence. In Proceedings of the 2022 Annual ACM-SIAM Symposium on Discrete Algorithms (SODA) (pp. 3732-3753). Society for Industrial and Applied Mathematics.

[Minor comments]

I believe improvements to certain notations and abbreviations would benefit many readers. While not limited to these examples, some are listed below. Naturally, space constraints in the main text may make it difficult to supplement explanations for all abbreviations or self-explanatory notations. However, should this paper be accepted, I would greatly appreciate your consideration of utilizing some additional space to address these points.

- I believe BBP in “BBP phase transition” stands for the discoverer's name, but for readers unfamiliar with this term, explicitly stating it upon first mention would make the text easier to read.

- I believe nnz(v) denotes the number of nonzero elements. Clarifying this upon first use would help avoid unnecessary reader confusion or concern.

- In Equation (1), |supp(v)| is used instead of nnz(v). Within the same context, using a consistent notation would prevent unnecessary confusion.

- The procedure in line 4 of Algorithm 1 is somewhat unclear.

Indeed, Section 2 of this paper provides detailed preparatory notation. However, my impression is that it feels somewhat incomplete. Specifically, when reading sequentially from the introduction, I felt the progression was not entirely smooth.

**Limitations:**

Section 3.2 clearly provides an interesting future direction. Moreover, the authors specify an open issue (Lines 153-156).

**Strengths And Weaknesses:**

[Strengths]

- *Significance*: This paper provides a simple yet non-trivial observation regarding Sparse PCA. It identifies cases where well-known algorithms (primarily for Model 1 in this paper) fail under the general model (Model 2 in this paper).

- *Significance*: This paper arrives at a definitive answer to a long-standing unsolved problem in Sparse PCA. More specifically, it provides a highly efficient solution algorithm with theoretical guarantees for the general model (Model 2).

- *Originality*: The semi-random model strategy employed to achieve this solution appears to be a non-trivial and novel tool in this context. It will likely offer significant benefits as a new perspective for subsequent research.

[Weaknesses]

- *Presentation*: While Section 1 provides a detailed overview of the research idea, catering to a diverse audience, the explanation of the actual main results in Section 3 is quite technical. It is not easy to pinpoint exactly where the new ideas emerge. This may not be a major issue for specialists deeply focused on this topic, but it might be somewhat difficult for researchers in adjacent fields to follow. Adding an intuitive overview of the main results in Section 3 (ideally accompanied by intuitive diagrams), strongly linked to the overview in Section 1, would likely increase the number of readers who can accurately grasp the paper's value.

---

> ### Author Rebuttal · Authors · 2026-03-31
>
> We thank the reviewer for their positive and encouraging comments about the importance and significance of our work. We address the primary questions raised by the reviewer below.
>
> **Improved sample complexity**: Although not mentioned by the reviewer, we are excited to announce that since submitting our original manuscript, we have been able to further improve the sample complexity in Theorem 2 from $n = \Omega(s^3 \log d)$ to $n = \Omega(s^2 \log d)$, matching the optimal sample complexity at the computational-statistical threshold. The improvement comes from a novel normalized potential function and bilinear form concentration, which bypasses the operator norm bottleneck in our original analysis. We will include this improved result in our revised manuscript. Please see the response to Reviewer mDvb for further details.
>
> **Presentation of the Proof**: We thank the reviewer for raising their helpful suggestions. To make the technical results in Section 3 more accessible to a broader audience, we will restructure this section and provide an overview at the beginning of the section. For the convenience of the reviewer, we also provide a roadmap below.
>
> The core of the proof is to show the potential function $\phi_t$ (the correlation between the current iterate and the top sparse eigenspace) increases from $\Omega(1/\sqrt{s})$ to $\Omega(1 - \Delta)$. This progress is divided into two phases depending on the value of $\phi_t$. The first phase covers $\phi_t \in [O(1/\sqrt{k}), 1/4]$: in each iteration, the power method step (Line of Algorithm 1) increases $\phi_t$ proportionally (Lemma 5), but suffers from subtractive error terms, including a sampling error (Corollary 1) and a truncation error (Lemma 2 and 3). We establish Lemma 6 by showing that the net progress remains strictly positive. The second phase covers $\phi_t \in [1/4, 1-\Delta]$ and is established in Lemma 7. Using a similar analysis to Phase 1, it bounds the errors in this high correlation regime to guarantee $\phi_t$ successfully reaches the target accuracy of $1 - \Delta$.
>
> **Regarding complementary venues**: We appreciate the reviewer's recognition of the theoretical depth of our contribution. We believe ICML is an excellent fit, as sparse PCA sits at the intersection of high-dimensional statistics and practical machine learning, and our experimental evaluation on real-world data highlights this applied relevance. That said, we are grateful for the suggestion of STOC/FOCS as complementary venues and would take that into sincere consideration.
>
> **Subgaussian assumption**: We thank the reviewer for this insightful observation. We chose to state our results under the subgaussian assumption for ease of exposition, as subgaussianity is a standard assumption for sparse PCA in the literature. As pointed out by the reviewer, since our core analysis relies critically on the matrix concentration inequalities in Fact 2 and Fact 3, we agree that similar results can be obtained for any distribution that guarantees a similar concentration bound. We will add a remark to clarify this point in the revised manuscript.
>
> We appreciate the reviewer’s careful assessment. We are encouraged that the reviewer found the main result important and the overall proof strategy convincing, and we hope the above clarifications make the remaining technical steps easier to verify.
>
> **Regarding Minor comments**:
>
> - **General Notation and abbreviations:** We will do a thorough pass of the whole article and ensure all abbreviations and symbols are defined when they are first mentioned.
>
> - **BBP Phase Transition:** BBP refers to the phase transition in random matrix theory, first introduced in a paper by Baik, Ben Arous, and Péché in 2005. We will expand this abbreviation in the manuscript.
>
> - **$\text{nnz}(v)$ vs $|\text{supp}(v)|$:** We will change all $|\text{supp}(v)|$ to $\text{nnz}(v)$ for notational consistency.
>
> - **Line 4 in Algorithm 1:** The algorithm restarts $d$-times with the standard basis vectors in $\mathbb{R}^d$. In the $i$-th outer loop, we initialize the power method with $\mathbf{e}_i$.

---

> > ### Author Rebuttal · Reviewer_Ha7F · 2026-04-03
> >
> > I am deeply grateful for the authors’ concise yet thorough response. All of my questions and concerns have been addressed. In particular, my understanding of the correctness of the paper’s theoretical contributions has improved significantly. While the paper includes some experimental verification, its primary contribution is based on theoretical rigor. Unfortunately, as I am not a theoretical expert in this field, my understanding of the correctness of Lemmats 5 through 7 was quite vague during my initial review. However, thanks to the authors’ response, I have gained a much clearer understanding of the correctness of the sequence of steps leading from their intuitive sketches to Theorem 2 (though I have not yet verified the coefficients and degrees in the detailed algebraic manipulations). I am grateful for the authors’ assistance.
> >
> > I have just one final comment for the authors. In their rebuttal, the authors reported “improvements regarding sample complexity.” Congratulations. I am very pleased to see this report. However, regarding whether or not to include this as a primary contribution to this paper, I would like to ask the authors to proceed with caution. More specifically, I would like them to be cautious about including results generated after the initial submission (i.e., results not included in the first draft) in the revised version. This is for the following reasons.
> >
> > - Including results generated after the initial submission as a major contribution seems unfair given that ICML submissions have a fixed common submission deadline.
> >
> > - Another reason is that, since ICML does not follow a review process involving repeated revisions and reviews, it is difficult to verify the accuracy of the updated content.
> >
> > If this paper is accepted, and unless the meta-reviewer (Area Chair) provides permission or instructions regarding revisions for the new results, I ask that you carefully consider, as the author’s responsibility, whether to include them as a primary contribution. For example, for the ICML 2026 paper, the authors could limit it to the results from the initial draft and submit the improved results as an extension to a journal (such as JMLR or TMLR).
> >
> > I apologize for making such a request. Once again, thank you for sharing this very interesting paper. And congratulations on producing even more fascinating results.

---

> > > ### Author Response · Authors · 2026-04-07
> > >
> > > We sincerely thank the reviewer for the thoughtful and generous follow-up. We are especially grateful that our rebuttal helped clarify the proof strategy and improve the reviewer's confidence in the theoretical contributions. We also appreciate the reviewer's careful point regarding post-submission improvements. We fully understand this concern, and if the paper is accepted, we will be careful to follow ICML's revision policies and the guidance of the AC regarding the scope of any updates included in the final version. Thank you again for your time, care, and encouragement.

---

### Decision · Program_Chairs · 2026-04-30

**Decision:**

Accept (regular)

**Comment:**

The paper studies sparse PCA from theoretical perspective. It is observed that under the non-spiked model, iterative algorithms (which are also called combinatorial algorithms by authors) provably fail. Authors then provide a new analysis of truncated power method that shows it recovers the sparse subspace in the new regime. All reviewers are positive. Concerns were addressed fully in rebuttal.